# LIS1 RNA-binding orchestrates the mechanosensitive properties of embryonic stem cells in AGO2-dependent and independent ways

Aditya Kshirsagar [1], Svetlana Maslov Doroshev[1], Anna Gorelik[1], Tsviya Olender[1], Tamar Sapir[1], Daisuke Tsuboi[2], Irit Rosenhek-Goldian [3], Sergey Malitsky [4], Maxim Itkin [4], Amir Argoetti[5], Yael Mandel-Gutfreund[5], Sidney R. Cohen [3], Jacob H. Hanna [1], Igor Ulitsky [6], Kozo Kaibuchi[2] & Orly Reiner[1] ✉

*Lissencephaly-1* (*LIS1*) is associated with neurodevelopmental diseases and is known to regulate the molecular motor cytoplasmic dynein activity. Here we show that LIS1 is essential for the viability of mouse embryonic stem cells (mESCs), and it governs the physical properties of these cells. LIS1 dosage substantially affects gene expression, and we uncovered an unexpected interaction of LIS1 with RNA and RNA-binding proteins, most prominently the Argonaute complex. We demonstrate that LIS1 overexpression partially rescued the extracellular matrix (ECM) expression and mechanosensitive genes conferring stiffness to Argonaute null mESCs. Collectively, our data transforms the current perspective on the roles of LIS1 in post-transcriptional regulation underlying development and mechanosensitive processes.

Lissencephaly-1 (*LIS1*) was the first gene identified as involved in a neuronal migration disorder[1]. Proper expression levels of the LIS1 protein are critical for both mouse and human brain development, with either decreased or increased expression affecting the developmental process[2–4]. The elimination of LIS1 is lethal during early development in mice and flies[2,4,5]. LIS1 is known to play a critical role in both neuronal and hematopoietic stem cells[6–14]. To date, these crucial roles of the LIS1 protein have been mainly attributed to its physical interaction with cytoplasmic dynein, which has been conserved throughout evolution[15–17]. The direct binding of LIS1 to dynein and additional accessory proteins results in conformational changes and modified mechanochemical properties of the molecular motor[18–21]. The interaction between LIS1 and cytoplasmic dynein impacts the many processes in which cytoplasmic dynein is involved, such as mitosis, interkinetic nuclear motility, neuronal migration, intracellular transport, and neuronal degeneration[4,10,13,16,22–25].

LIS1 has also been implicated in additional activities unrelated to its interactions with the molecular motor. LIS1 affects the cytoskeleton by modulating microtubules and the actin mesh[22,26–29]. LIS1 localizes in the nucleus, where it interacts and affects the activity of MeCP2[30]. In human embryonic and neuronal stem cells, LIS1 affects gene expression and the physical properties of the colonies[10].

Here we investigate the dosage-related roles of LIS1 in embryonic stem cells using a multidisciplinary approach and detect novel and unexpected functions for LIS1 in post-transcriptional regulation and affecting the physical properties of mESCs. We found that LIS1 binds RNA and interacts with numerous proteins, many of which are RNA-binding proteins (RBPs), including some belonging to the Argonaute complex. LIS1 is mainly bound to the nascent RNA of protein-coding genes, and the number of LIS1- binding sites within introns were negatively correlated with intron splicing efficiency. A different outcome was noted when LIS1 bound to microRNA (miRs), which coincided with their increased expression. Overexpression of LIS1 in the absence of AGO1-4 enabled low but significant expression of a subset of miRs.

[1]Departments of Molecular Genetics and Molecular Neuroscience, Weizmann Institute of Science, Rehovot, Israel. [2]International Center for Brain Science, Fujita Health University, Toyoake, Japan. [3]Department of Chemical Research Support, Weizmann Institute of Science, Rehovot, Israel. [4]Department of Life Sciences Core Facilities, Weizmann Institute of Science, Rehovot, Israel. [5]Faculty of Biology, Technion-Israel Institute of Technology, Haifa, Israel. [6]Department of Immunology and Regenerative Biology, Weizmann Institute of Science, Rehovot, Israel. ✉e-mail: orly.reiner@weizmann.ac.il

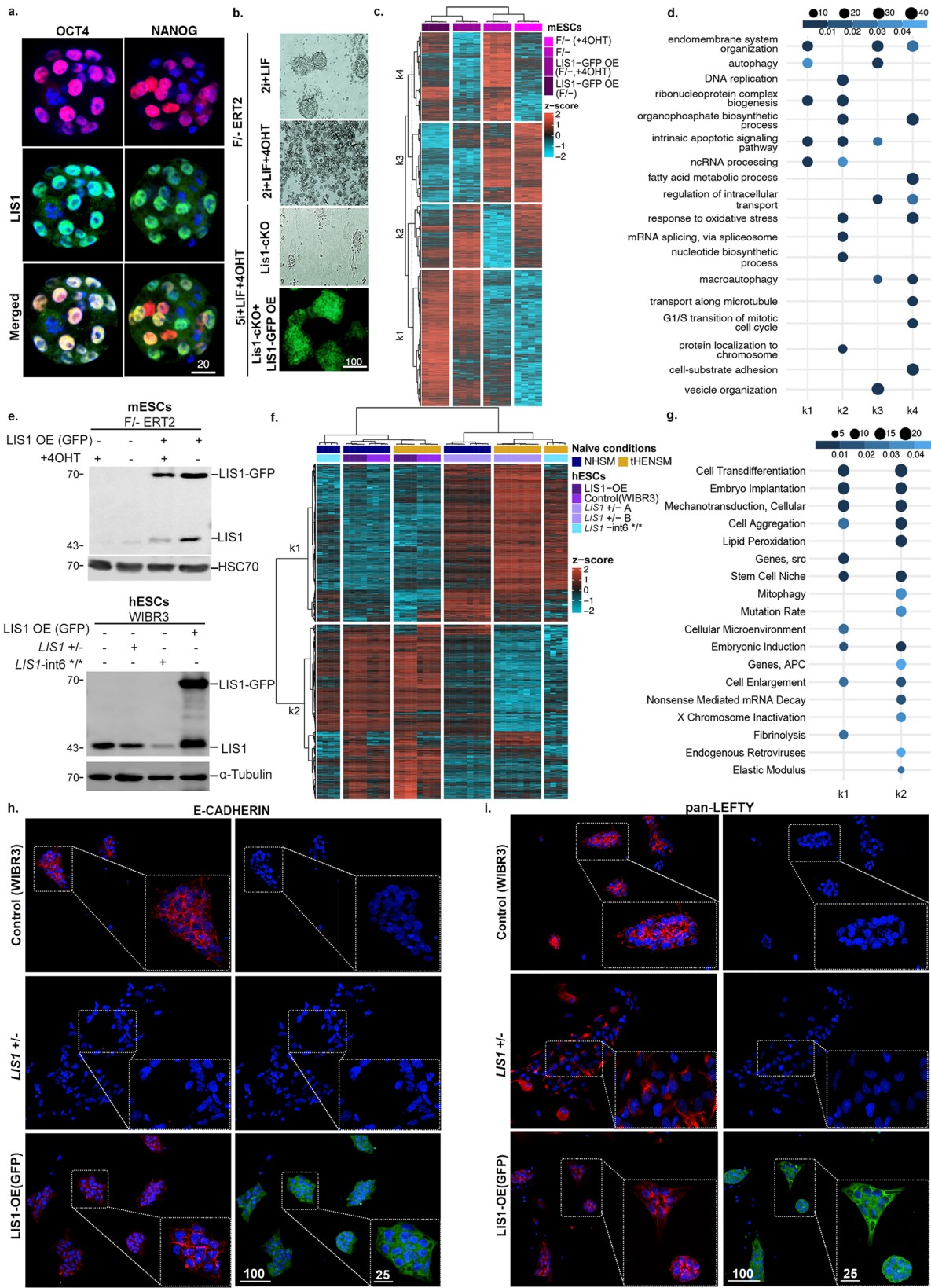

Whereas AGO1-4 KO cells were soft, LIS1 overexpression changed the expression of genes related to the extracellular matrix and mechanosensitivity and resulting in a robust increase in the stiffness of mESCs lacking Argonaute proteins. Collectively, these findings necessitate re-evaluating the roles of LIS1 during development.

## Results

### Dosage-sensitive effects of LIS1 in pluripotency networks

*Lis1-/-* mice are early embryonic lethal[2,4,5]; therefore, we examined the localization of LIS1 in E3.5 pre-implantation embryos. LIS1 was detected in the cytoplasm as well as the nucleus, where it partially colocalizes with either OCT4 or NANOG (Fig. 1a), suggesting unknown

**Fig. 1 | LIS1 dosage affects gene expression. a** LIS1 colocalizes with NANOG and OCT4 in the nucleus. Pre-implantation embryos (*n* = 3) from embryonic day 3.5 were immunostained with anti-NANOG, anti-OCT4, and anti-LIS1 antibodies (scale bar, 20 μm). **b** Phase-contrast and fluorescent images of *Lis1* F/- ERT2 mESCs cultured in 2iL media with or without 4-OHT. Lis1 F/- ERT2 and mESCs overexpressing LIS-GFP treated with 4-OHT using alternative naïve (5iL) media conditions. Images are representative of at least two independent experiments (scale bar,100 μm). **c** A heatmap of 2654 differentially expressed genes across samples with different *Lis1* gene dosages in *Lis1* F/- ERT2 (F/- ERT) derived mESCs cultured in 5i+LIF media. The data are shown on a Z-score scale of the variance stabilizing transformation on normalized reads. **d** Analysis of gene set over-representation test for four clusters obtained from the k-means clustering (k1, k2, k3, and k4) shows that different levels of LIS1 modulate the expression of genes enriched in specific Gene Ontology Biological Process (GO-BP) terms. **e** LIS1 levels in mouse (upper panel) and human lines

(lower panel). Mouse lines: LIS1-GFP overexpression (OE) on the background of F/-ERT2, before and after 4-Hydroxytamoxifen (4OHT) addition (*n* = 3). Lower panel: LIS1-GFP over expression (OE), *LIS1* heterozygous (+/−) and lissencephaly-associated intronic mutation *LIS1*-int6*/* (*n* = 3). **f**) A heatmap of 927 differentially expressed genes across samples with different *LIS1* gene dosages in hESCs Samples come from two independent media conditions; human naïve, NHSM, and tHENSM. The data are shown on a Z-score scale of the variance stabilizing transformation on normalized reads. **g** Gene set over-representation test analysis showing the comparison of dose-dependent upregulated and downregulated k-means clusters (k1 and k2) in hESCs enriched for Medical Subject Headings (MeSH) terms. **h, i** Representative immunofluorescence images of immunostainings conducted in WIBR3 (control), *LIS1* +/−, and LIS1-OE (GFP) cultured in tHENSM media (*n* = 3) using anti- E-CADHERIN, and pan-LEFTY (LEFTY-A and LEFTY-B) antibodies, respectively (scale bar,100 μm. Inset represents 2.5x zoom; scale bar 25 μm).

nuclear functions. The transcription factors OCT4 and NANOG regulate the expression of the dynamic transcription network and are required for the pluripotency and proliferation of embryonic stem cells (ESCs)[31–34]. So far, whether *Lis1-/-* mESCs are viable has not been established. To generate null cells, *Lis1 floxed/-:Cre-ERT2 (Lis1 F/-:ERT2)*, mESC lines were derived from blastocysts, and the *Lis1* deletion was induced by tamoxifen (4-OHT) treatment. When *Lis1* was deleted in the presence of the standard 2i+LIF media, the cells died rapidly (Fig. 1b). We then reasoned that 5i+LIF media, modified from human naive media[35], may support cell viability (Supplementary Table 1). mESC colonies containing the Oct4 reporter line[36] appeared compact and pluripotent in this media compared to 2i+LIF, or FBS + LIF (Supplementary Fig. 1a).

The necessity to remove the Wnt inhibitor (XAV939) in the 5i+LIF media was suggested due to the findings that reduced LIS1 dosage in brain organoids resulted in inhibition of the Wnt pathway[37], and was supported by examining gene expression (Supplementary Fig. 1b). The 5i+LIF modified media-enabled three passages of cultured *Lis1*-deleted cells, after which the cells died (Fig. 1b). These findings demonstrated that LIS1 expression is required for the viability of mESCs. RNA-expression data was obtained from the different LIS1-dosage-dependent genotypes (Fig. 1c). When comparing the differentially expressed (DE) genes between the highest and the lowest LIS1 dosage (LIS1-GFP OE, floxed/- background versus floxed/- treated with tamoxifen, that is *Lis1-/-*, respectively), 2150 genes were upregulated, and 904 were downregulated (Supplementary Data 1). The DE genes came together in four k-means clusters (Fig. 1c). Analysis of enriched Gene Ontology Biological Process (GO-BP) terms showed that LIS1 dosage affects many fundamental processes, including DNA replication, mitosis, apoptosis, and autophagy. RNA-related functions such as biogenesis, splicing, and noncoding RNA processing were also enriched in this analysis. LIS1 is also involved in biosynthetic and metabolic processes, including some pertaining to nucleotides, fatty acids, and organophosphate (Fig. 1d). The levels of LIS1 in different lines were examined and quantified, and significantly, the ectopic expression of LIS1-GFP rescued the lethality of *Lis1* deletion (Fig. 1b, e, Supplementary Fig. 1c).

We derived additional F/-, WT, and LIS1-DsRED overexpression (OE) mESC lines from blastocysts and immunostained for OCT4. We observed that LIS1 OE reduced the presence of OCT4 in the nucleus (Supplementary Fig. 1d, e). Metabolomic analysis of mESCs revealed changes in nucleotides such as deoxyuridine, deoxycytidine monophosphate, and dAMP, different levels of amino acids such as proline and aspartate, and changes in essential metabolism molecules such as nicotinamide adenine dinucleotide (NAD). The fatty acid metabolic profile showed changes in monounsaturated fatty acids such as erucic acid-like and polyunsaturated fatty acids such as eicosapentaenoic acid (EPA) and docosahexaenoic acid (DHA) (Supplementary Fig. 2a–c significant changes are shown in heatmaps related to Supplementary Data 2). Moreover, genes affected by LIS1 expression are involved in vesicle organization, intracellular transport, and

cell-substrate adhesion, all of which could be associated with known LIS1 functions (Fig. 1d).

Next, we examined the role of LIS1 expression in regulating pluripotency in human ESCs. We used two media conditions[35] and five isogenic lines; the control wild-type, two previously published *LIS1*+/−lines[10], LIS1 overexpression (OE), and a lissencephaly-associated intronic mutation in intron 6 affecting splicing[38] (*LIS1-int6*/*) in the homozygous form, that slightly reduced LIS1 levels (Fig. 1e, Supplementary Fig. 3a−c). RNA-seq revealed that the overexpression and wild-type were detected in one group, and *LIS1-int6*/* clustered with the *LIS1*+/− lines only in one growth condition (Fig. 1f). A total of 927 DE genes between the wild-type and *LIS1*+/− lines were noted (Supplementary Data 3a). The selected Medical Subject Headings (MeSH) terms indicated changes in the stemness and differentiation potential, RNA, and the physical properties of the cells, such as mechanotransduction and elastic modulus (Fig. 1g). Gene Analytics pathway analysis revealed genes involved in the extracellular matrix organization, pluripotency, and others (Supplementary Data 3b, Supplementary Fig. 3d). A subset of the differentially expressed genes was common to mice and humans (Supplementary Data 3c). We confirmed the differential expression of a few of the genes by immunostaining. The pluripotent transcription factors OCT4 and NANOG levels were increased in cells with higher LIS1 dosage, with the highest in LIS1-OE hESC (Supplementary Fig. 3c, Statistics in Supplementary Data 6a). E-cadherin was particularly interesting with its central role in embryonic stem cell pluripotency, epithelial-to-mesenchymal transition (EMT), and mechanotransduction[39–41]. *LIS1*+/− cells did not express E-cadherin. However, the control and the LIS1-OE cells expressed it at high levels (Fig. 1h, Supplementary Fig. 4a). LEFTY, a member of the TGF-beta family, contributes to the remodeling of the extracellular matrix and regulates actin polymerization and stiffness[42–44]. The wild-type and LIS1-OE ESC colonies expressed LEFTY in a polarized manner. In contrast, there was a reduced expression in the *LIS1*+/− cells (Fig. 1i). We examined the localization and expression of β-Catenin, the primary downstream target of the Wnt pathway, in hESCs and found the most striking difference in *LIS1*+/− colonies, where the protein appeared mislocalized (Supplementary Fig. 4b−e). Collectively, our data demonstrate that LIS1 is expressed in the nucleus and the cytoplasm and is essential for the pluripotency and survival of embryonic stem cells. Although our custom-designed media-enabled three cell passages, the embryonic stem cells died. LIS1 levels affected gene expression in a dosage-specific manner and were found to be involved in multiple basic cell biological processes. We hypothesized that the known interactome of LIS1, composed of mainly cytoskeleton-related proteins, was insufficient to explain all of these changes, and we proceeded to identify novel LIS1-interacting proteins.

## The LIS1 interactome

The known LIS1 interactome is composed of 148 proteins, many of which are involved in microtubule-based processes (BioGRID[45] and

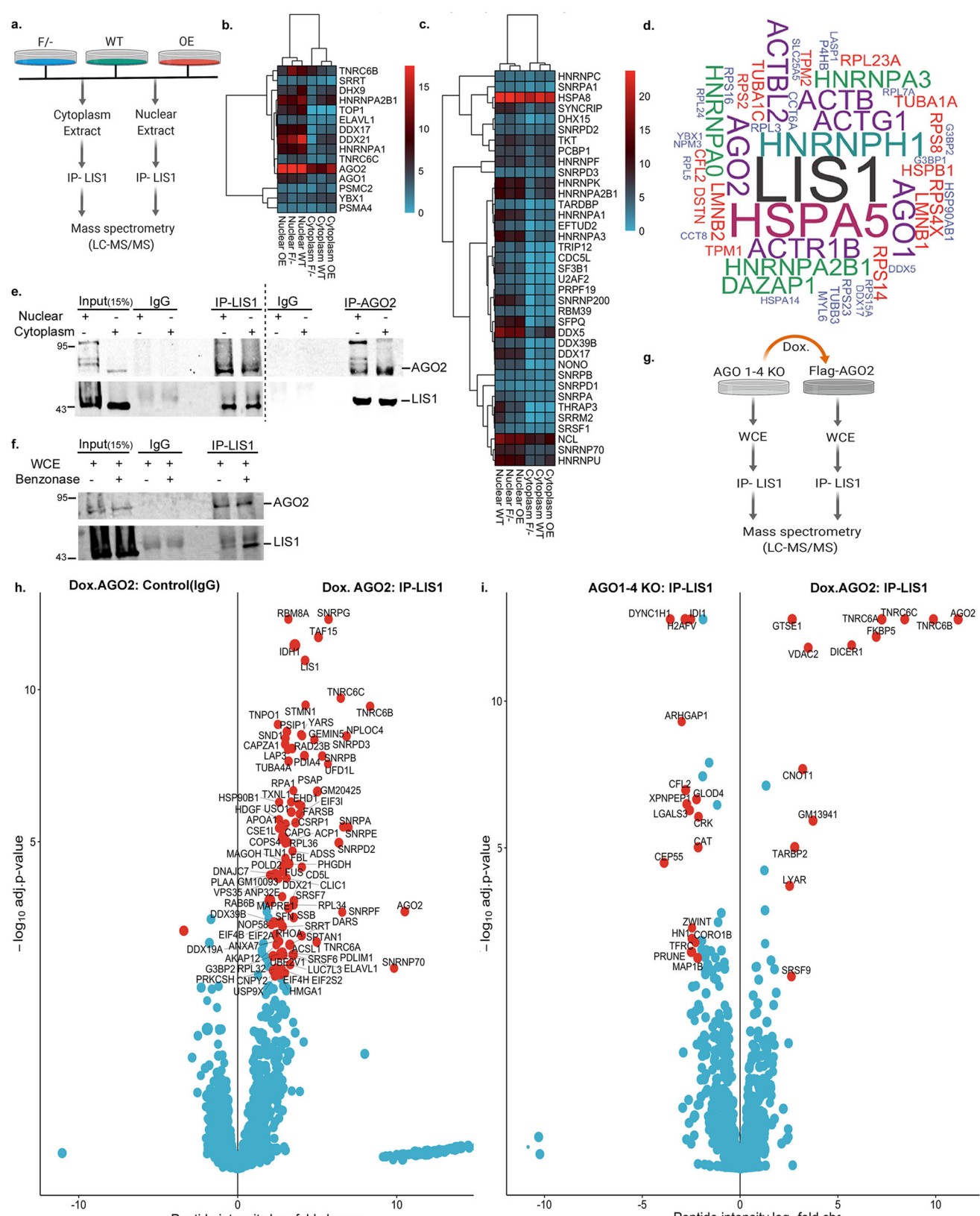

STRING[46] databases), and very few are nuclear proteins. Novel LIS1-interacting proteins were identified by immunoprecipitation of LIS1 from nuclear or cytoplasmic fractions of mESCs expressing graded levels of LIS1 followed by mass-spectrometry (LC-MS/MS) (Fig. 2a, Supplementary Fig. 5a–d). A total of 726 unique proteins were identified, many of which were unknown to complex with LIS1

(Supplementary Data 4a, Supplementary Fig. 5e, f). More than half of these proteins (385) were designated in the RBPbase (https://rbpbase. shiny.embl.de/) as "RNA binding-GO-Mm". Most of the LIS1 inter-actome was shared in the cells with different LIS1 expression levels (WT, F/-, and OE) (Supplementary Fig. 5e). The top consistent and genotype-specific GO terms related to mRNA splicing and cellular

**Fig. 2 | Novel LIS1-interacting proteins include a repertoire of RNA-binding proteins. a** A schematic illustration of the strategy used to identify LIS1-interacting proteins in mESCs. The cytoplasmic and the nuclear fractions from each genotype (F/-; Floxed/-, hypomorph allele, WT; Wild Type, and OE; LIS1-dsRED over-expression) were separated. LIS1 was immunoprecipitated (IP) using anti-LIS1 antibodies followed by mass spectrometry (total $n = 4$ for each genotype). **b** A heatmap of the RISC complex and P-body proteins derived from the mass spectrometry results. **c** A heatmap of splicing factors and nuclear speck proteins identified as significant LIS1 interactors. For (**b**, **c**), the scale represents razor and unique peptides detected averaged across replicates for each protein in each fraction per genotype. **d** The top 50 LIS1 interactors in *Drosophila melanogaster*. The size of the protein name corresponds to the total number of peptides obtained in LIS1 IP for each mouse orthologue. **e** Validation of the interaction between endogenous LIS1 and AGO2 in the nuclear and cytoplasmic fractions in hESCs as determined by coimmunoprecipitation followed by western blotting. **f** The interaction between LIS1 and AGO2 is maintained independent of RNA. Total whole cell extract used for immunoprecipitation was treated with Benzonase to ensure the complete absence of RNA. **e**, **f** The immunoprecipitations were performed with the indicated antibodies, and IgG was used as a control for the IP with the input (15%) as shown. **g** Schematic illustration of the strategy to identify AGO2-independent LIS1 interacting proteins in AGO1-4 KO (AGO1/2/3/4 knockout) and AGO2-dependent (doxycycline-inducible AGO2 in AGO1-4 KO; Dox.AGO2) mESCs. LIS1 IP in combined nuclear and cytoplasmic fractions was performed in AGO1-4 KO and Dox.AGO2 followed by mass spectrometry ($n = 4$). **h** A volcano plot for the ratios of peptide intensities of proteins detected with mass spectrometry in LIS1 IP versus control (IgG, nonspecific peptides) in Dox. AGO2AGO1-4KO. **i** A volcano plot for the ratios of peptide intensities of proteins detected in mass spectrometry with LIS1 IP in AGO1-4 KO and Dox.AGO2, respectively. For (**h**, **i**), limma-generated log2 fold-change and adjusted *p*-values for multiple comparisons from the DEP package[109] were used. The proteins with a log2 foldchange ≥ 2 and an adjusted *p*-value ≤ 0.01 were considered significant and are highlighted in red.

stress response (Supplementary Fig. 5f). Many LIS1-interacting proteins belonged to the RISC complex and P-bodies, interacting with AGO2 (Fig. 2b). A second prominent group of proteins was composed of RNA-binding proteins functioning in other activities, including alternative splicing, mRNA stabilization, RNA metabolism, transcriptional and translational regulation (Fig. 2c). We found several members of the heterogeneous nuclear ribonucleoproteins (hnRNPs)[47], small nuclear ribonucleoproteins (snRNPs)[48], DDX, and DHX gene families[49]. A subset of these two groups of proteins is shown in the heatmaps, demonstrating that the protein-protein interactions are enriched in the nucleus and occur in cells with graded levels of LIS1. We composed a list of the LIS1 interactome by combining our data with BioGRID data resulting in 1274 unique proteins (Supplementary Data 4a–c).

To assess the conservation of the identified LIS1-interactome, we compared our results with a previously published large-scale *Drosophila* interactome[50]. A significant number of the translated *Drosophila* orthologs overlapped with the LIS1-interacting proteins in mESCs (Supplementary Fig. 5g). These included members of the AGO clade, heat-shock, chaperone proteins, and several splicing factors, suggesting that these interactions are evolutionarily conserved (Fig. 2d). We further detected a significant overlap between the nuclear interactome of LIS1 and AGO2 and indicated common and distinct LIS1 and/or AGO2 processes[51] (Supplementary Fig. 5h, i). For example, we detected "transcription repression" and "extracellular exosome" among the common GO terms.

Following this significant overlap and the findings that mutations in *AGO2* result in intellectual disability and developmental delay[52], we reasoned that at least part of LIS1 functions might be mediated through its interaction with AGO2. Nevertheless, we note significant overlaps between the LIS1 interactome, the interactomes of thirty-one RBPs, and sixteen LIS1 interacting heat shock proteins (depicted in Fig. 2c and Supplementary Data 4d, e). We confirmed the interaction between LIS1 and AGO2 by co-immunoprecipitation followed by western blot in the nuclear and cytoplasmic fractions (Fig. 2e). We treated protein lysates with benzonase nuclease, and observed that LIS1 immunoprecipitated AGO2, suggesting that LIS1 and AGO2 interact regardless of the presence of RNA in the complex (Fig. 2f).

To systematically identify the Argonaute-dependent LIS1 protein interactors, we used an *Ago1,2,3,4* knockout line with the doxycycline-dependent expression of human AGO2, with and without the presence of doxycycline, to perform LIS1-IP followed by mass-spectrometry[53,54] (Fig. 2g and Supplementary Data 4b). In the presence of AGO2, LIS1 bound to multiple interacting proteins (including LIS1 itself) compared with control IgG, supporting the LIS1-interactome identified above (Fig. 2h). In the absence of the Argonaute proteins, LIS1 complexed with several known LIS1-interacting proteins, such as cytoplasmic dynein heavy chain, the microtubule-associated protein MAP1B, and additional proteins, some of which have not been previously reported

to interact with LIS1, such as centrosomal protein CEP55, kinetochore protein ZWINT, and actin regulator, cofilin-2 (CFL2) (Fig. 2i, left side). The induction of AGO2 expression (Dox.AGO2, Fig. 2i, right side) resulted in additional protein interactions. These proteins included AGO2, TNRC6A-C, DICER1, FKBP5, TARBP2, and CNOT1, all of which were known to interact with AGO2. Some novel AGO2-interacting proteins were identified, including the voltage-dependent anion channel pore-forming protein VDAC2, GTSE1, which may be involved in the p53-induced cell cycle arrest, and the LYAR protein involved in the processing of pre-rRNAs.

## LIS1 is an RNA-binding protein

As we have shown that most of the LIS1 interactome is composed of RNA-binding proteins (RBPs), we set out to map the extent of LIS1's interaction with RNA. Furthermore, a previous high-throughput study identified LIS1 as an RBP[55]. The interaction of LIS1 with RNA mapped to amino acids 72-88 (LNEAKEEFTSGGPLGQK), which is contained within the N-terminal domain of LIS1 and is not part of the WD repeats. To examine LIS1-RNA interaction, we conducted LIS1 single-end enhanced crosslinking immunoprecipitation (seCLIP) experiments (Supplementary Fig. 6a)[56]. The results indicated that LIS1 binds to multiple loci (40,165 peaks), mainly found within the introns of protein-coding RNA (87%, Fig. 3a). LIS1 is preferentially bound to RNA of highly expressed genes and those with relatively large introns (Supplementary Fig. 6b, c). LIS1 seCLIP sites were more frequent in the first two introns (Supplementary Fig. 6d). Metagene plot of the data indicated that LIS1 preferentially binds to introns within close proximity to the donor splice site (Fig. 3b). A pattern of such asymmetric binding was also detected in data derived from AGO2-eCLIP experiments in both control cells and even more so in cells lacking Dicer, an essential component of the RISC complex[57] (Supplementary Fig. 6e, f). In addition, we noted significant two-fold enrichment of the U1 snRNP binding site in the peaks, suggesting the LIS1 may be involved in splicing regulation. LIS1-binding consensus sequences in the eCLIP peaks were GU-rich, as could be expected from intronic- or close to-donor splice site sequences (Fig. 3c). Two DE genes that contain LIS1 binding sites in their introns are *Lefty2* (Fig. 3d), and the gene encoding E-cadherin, *Cdh1* (Fig. 3e), which is not expressed in the *LIS1*+/− hESCs (Fig. 1g). Considering the preferential binding of LIS1 in proximity to donor splice sites, we tested whether LIS1 RNA-binding affected splicing. To better understand the impact of LIS1-RNA binding, we sequenced *Lis1 F/-*, WT, and *LIS1 OE* mouse ESC lines generated from blastocysts. We detected a negative correlation between the number of LIS1 seCLIP sites and splicing efficiency in wild-type RNAseq data (Fig. 3f, green, Bulk RNA-Seq splicing data using MAJIQ and RMATS in Supplementary Data 5, and the statistics in Supplementary Data 6b). The most marked changes were noted when comparing *LIS1 OE* to *Lis1 F/-* (Supplementary Data 5a–c), and fewer events were noted when comparing the WT to *Lis1 F/-* (Supplementary

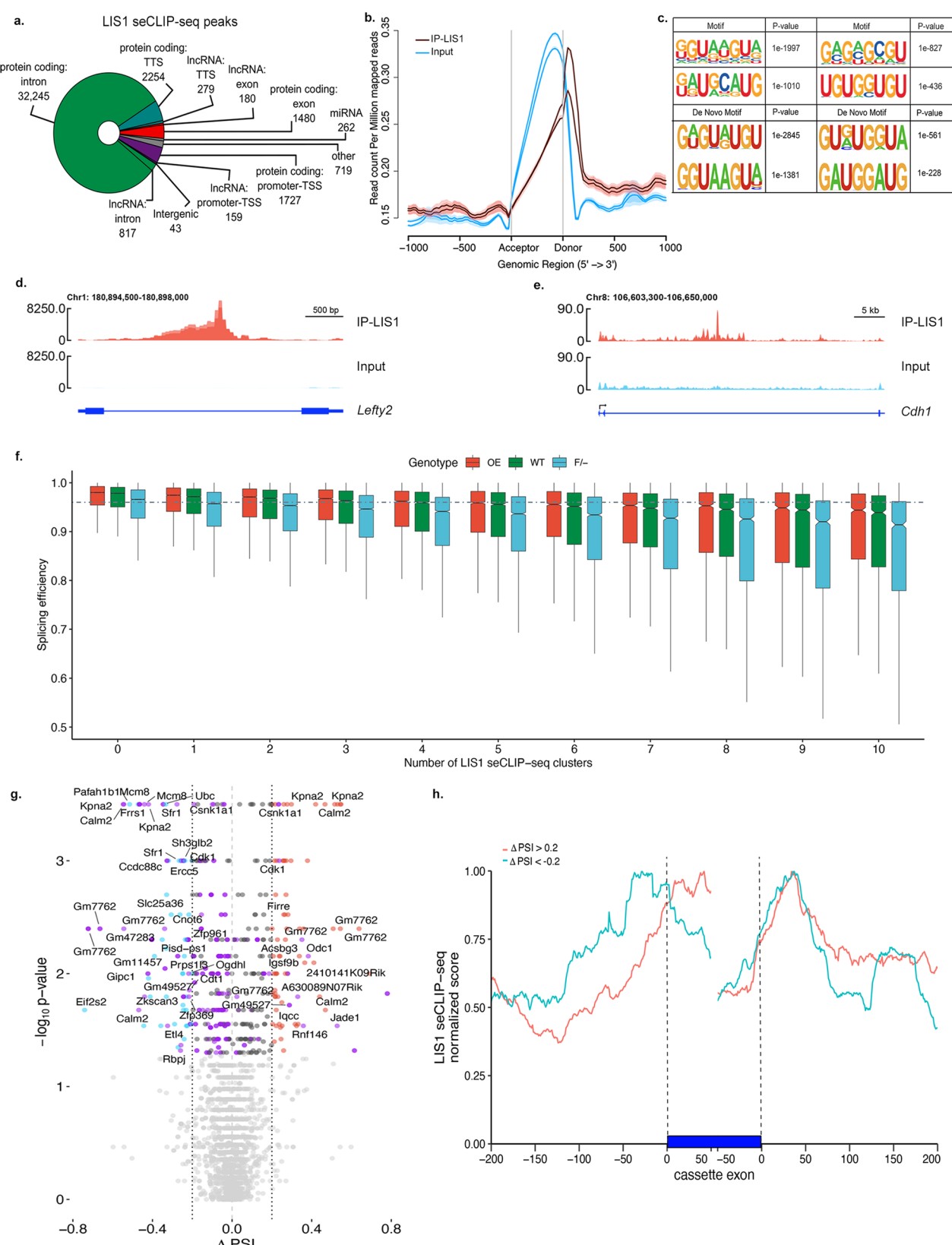

Data 5d–f). When LIS1 is decreased (as in the case of F/- cells, light blue), the splicing efficiency is progressively reduced. Conversely, the splicing efficiency was slightly but significantly increased upon LIS1 OE (Fig. 3f, red). We then examined local splicing variations (LSVs) by contrasting the LIS1 OE versus the F/- RNAseq data (Supplementary Fig. 7a). The majority of the cases using MAJIQ included cassette exon, alternative intron, and alternative first exon. In the rMATS analysis, skipped exon events were the most frequent. Most LSVs were annotated, yet many were novel (Supplementary Fig. 7b, c). Top Reactome terms included RHO GTPase cycle, organelle biogenesis and maintenance, centrosome maturation, and cilia assembly (Supplementary Fig. 7c). Examples for multiple LSVs are shown for the *Meg3* and *Rian* loci (Supplementary

**Fig. 3 | LIS1 RNA-binding properties and splicing regulation. a** LIS1 seCLIP-seq reproducible peaks (*n* = 40165) categorized by functional genomic regions. Peak regions identified by CLIPper peak caller with log2 fold change ≥ 3, adjusted *p*-value ≤ 0.001 were considered significant. **b** Coverage profile plot for the mean (±SEM) LIS1 seCLIP-seq reads distribution across exons. Brown lines represent two individual replicates for LIS1, and blue lines indicate the input (SMI; size-matched input). **c** Homer enriched motifs of LIS1 seCLIP peaks. Top and bottom are the four motifs enriched with the most significant *p*-values predicted by each of the homer known motif search and de novo motif search algorithms, respectively. **d**, **e** LIS1 eClip read coverage in *Lefty2* 1st intron (**d**.) and *E-cadherin* 2nd intron (**e**.) Red tracks- merged biological seCLIP-seq replicates (IP), Light blue tracks- input (SMI). **f** Intron level quantification of splicing efficiency in LIS1 Overexpressing line (LIS1 OE, red), Wild-type (WT, green), and *Lis1* F/- (F/-, blue) with respect to the LIS1 seCLIP-seq clusters (the experiment includes *n* = 4 RNA-seq replicates for each genotype). All the comparisons were significant across each genotype in each cluster group. Boxplots show median and lower or upper quartiles; whiskers show inner fences. Kruskal–Wallis test, followed by two-sided Dunn's all-paired test for multiple comparisons, *p*-values for all the comparisons are reported in extended Supplementary Data 6a–f. **g** A volcano plot of differentially retained introns between OE and F/- derived from MAJIQ alternative intron usage analysis. Significant upregulated events are in red, and (ΔPSI ≥ 0.2, *p*-value ≤ 0.05) and down-regulated events (ΔPSI ≤ −0.2, *p*-value ≤ 0.05) are in blue. Events with retained introns and *p*-value ≤ 0.05 are highlighted in purple. The remaining events with a *p*-value ≤ 0.05 are dark gray. Light gray indicates events with a *p*-value ≥ 0.05. Adjusted *p*-values from MAJIQ's two-sided Wilcoxon test were used to determine significance. Gene symbols for the de novo events are shown (*n* = 8 RNA-seq replicates for each genotype). **h** LIS1 seCLIP signal in regions with differentially spliced events annotated to cassette exons with a retained intron (MAJIQ ΔPSI ≥ 0.2). The red line indicates LSVs, which were found to be higher in LIS1OE, whereas the blue line indicates such events that are lower in LIS1 OE (red; *n* = 320, blue; *n* = 280).

Fig. 7d, e). Retained intron events were plotted, showing more events occurring in the F/- (Fig. 3g). These results indicate that when LIS1 levels are reduced, splicing efficiency decreases, increasing intron retention events. Conversely, in the case of LIS1 OE, it is possible to note fewer intron retention events (Fig. 3g), suggesting that the splicing out of introns from the nascent transcripts is more efficient. As shown above, elevated LIS1 levels result in more efficient splicing than F/- within the group of genes with the same LIS1 seCLIP binding sites (Fig. 3f). In addition, we wish to note that in the case of the LIS1 OE, additional LIS1 binding sites may not have been captured in our LIS1 seCLIP data set derived from WT cells. An example of differential splicing between the F/- and the OE is shown using Sashimi plots for the *Meg3* locus (Supplementary Fig. 7d). The position of LIS1 seCLIP sites in relation to cassette exons was plotted (Supplementary Data 7a, Fig. 3h). While a clear peak is seen next to the donor sequence (as depicted in Fig. 3b), we noted a difference between LSVs that were higher in OE, where the peak is in the exon and those that were lower in the OE, where the peak was in the intron close to the acceptor site (Fig. 3h). Part of the LIS1-bound genes was also DE (Supplementary Fig. 7f-h, Supplementary Fig. 8a, Supplementary Data 7b). Gene analytics pathway enrichment analysis showed that the top affected pathways included DNA Damage, Apoptotic Pathways in Synovial Fibroblasts, Gene Expression (Transcription), ERK Signaling, Nervous system development, Wnt / Hedgehog / Notch, Mesodermal Commitment Pathway, MiRNA Regulation of DNA Damage Response, and Transcriptional Regulation of Pluripotent Stem Cells (Supplementary Data 7c, Supplementary Fig. 8b). A complete list of transcripts showing significant changes in splicing efficiency and bound to LIS1 is outlined in Supplementary Data 7d. We further compared the list of genes that exhibited differential expression and differential splicing efficiency (OE and F/-) with the alternatively spliced genes detected by MAJIQ, resulting in 1251 genes (Supplementary data 7e). In addition, a subset of LIS1 bound DE genes exhibited differential accessibility as detected by Assay for Transposase-Accessible Chromatin using sequencing (ATAC)-seq in the different lines, including the *Meg3-Mirg* locus (Supplementary Fig. 8c) (details in methods, statistics in Supplementary Data 8). The MARS-seq data generated from the *Lis1-/-* cells (Fig. 1c) is incompatible with the splicing analysis. Nevertheless, we noted differential expression of a large cohort of RNA binding proteins, including splicing factors such as *RBFOX2, SNRNPA1, SNRNPD1, HNRNPC, HNRNPH3*, and *HNRNPM*, indicative of global changes in splicing. Our data indicate that LIS1 is an RBP that preferentially binds intronic sequences of highly expressed protein-coding genes near the splicing donor site. In addition, the increase in seCLIP LIS1 binding sites within a gene is negatively correlated with splicing efficiency, and increased expression of LIS1 improves the efficacy of RNA splicing.

## LIS1 affects the expression of miRs

We next proceeded to examine the effect of LIS1 on small RNA. Argonaute proteins are best known for their role in post-transcriptional regulation by microRNAs (miRs)[58]; therefore, we conducted RNA-seq and small-RNA seq from mESCs. We detected 85 DE mature miRs, most of which were upregulated by LIS1 overexpression (Fig. 4a, Supplementary Data 9). Another smaller group of miRs exhibited lower expression in the F/- compared to WT and OE. These miRs included miR302a-3p, miR302b-3p, miR302c-3p, miR302d-3p, miR302a-5p, miR-142a-3p, miR-335-3p, miR-335-5p, and miR-211-5p. Previous studies have shown that the miR302 cluster is important in mechanosensitivity, neural differentiation, and reprogramming[59-64]. Among the upregulated DE miRs, the *Meg3-Mirg* locus was highly represented. It included 76 small RNA genes (64 mature miRs) (Fig. 4b). Multiple LIS1 seCLIP peaks were detected in this locus, many of which were on top of or in close proximity to miRs (Fig. 4b). Not only was the expression of miRs in this locus significantly increased but also the expression of the protein-coding and non-protein-coding genes (Fig. 4c, Bulk RNA-Seq data in Supplementary Data 10, Supplementary Fig. 9a, b). ATAC-seq experiments revealed that this locus and the associated enhancer region are relatively open in the LIS1 OE, facilitating transcription (Fig. 4b). Most of the DE miRs were located in introns or in close proximity to protein-coding or noncoding genes, many of which were also DE (Fig. 4c). The expression of the majority of these genes was elevated in the LIS1 OE line (Fig. 4c). We further correlated the expression of miRs with known mRNA targets in our RNA-seq data (Supplementary Fig. 9c). We noted that the expression of miRs close to LIS1 seCLIP peaks is significantly higher than those located at a distance from the LIS1 seCLIP peaks (Fig. 4d, e). We then examined the expression of miRs in relation to LIS1 seCLIP peaks using a threshold of 2 kb (Fig. 4e, statistics in Supplementary Data 6c). The expression of a miR located in the LIS1 seCLIP site (±2 kb) was significantly higher than in all other categories. If the location of the miR was further away from the peak (>2 kb), their level of expression was dependent on whether they were in noncoding or protein-coding introns, with those located in introns of noncoding RNAs having higher expression than those found in protein-coding introns or in comparison to all other miRs (statistics in Supplementary Data 6, c).

Collectively, our data indicate that increased levels of LIS1 stimulate the expression of a subset of miRs. LIS1's binding to intronic sequences on top of or near miRs is correlated with higher expression of these miRs. Still, if the distance to the LIS1 seCLIP binding site is beyond 2 kb, the miRs located in introns of non-protein-coding genes will tend to be expressed. In contrast, those found in introns of protein-coding genes will likely not be expressed in the mESCs. LIS1 likely affects the expression of miRs at multiple levels[65].

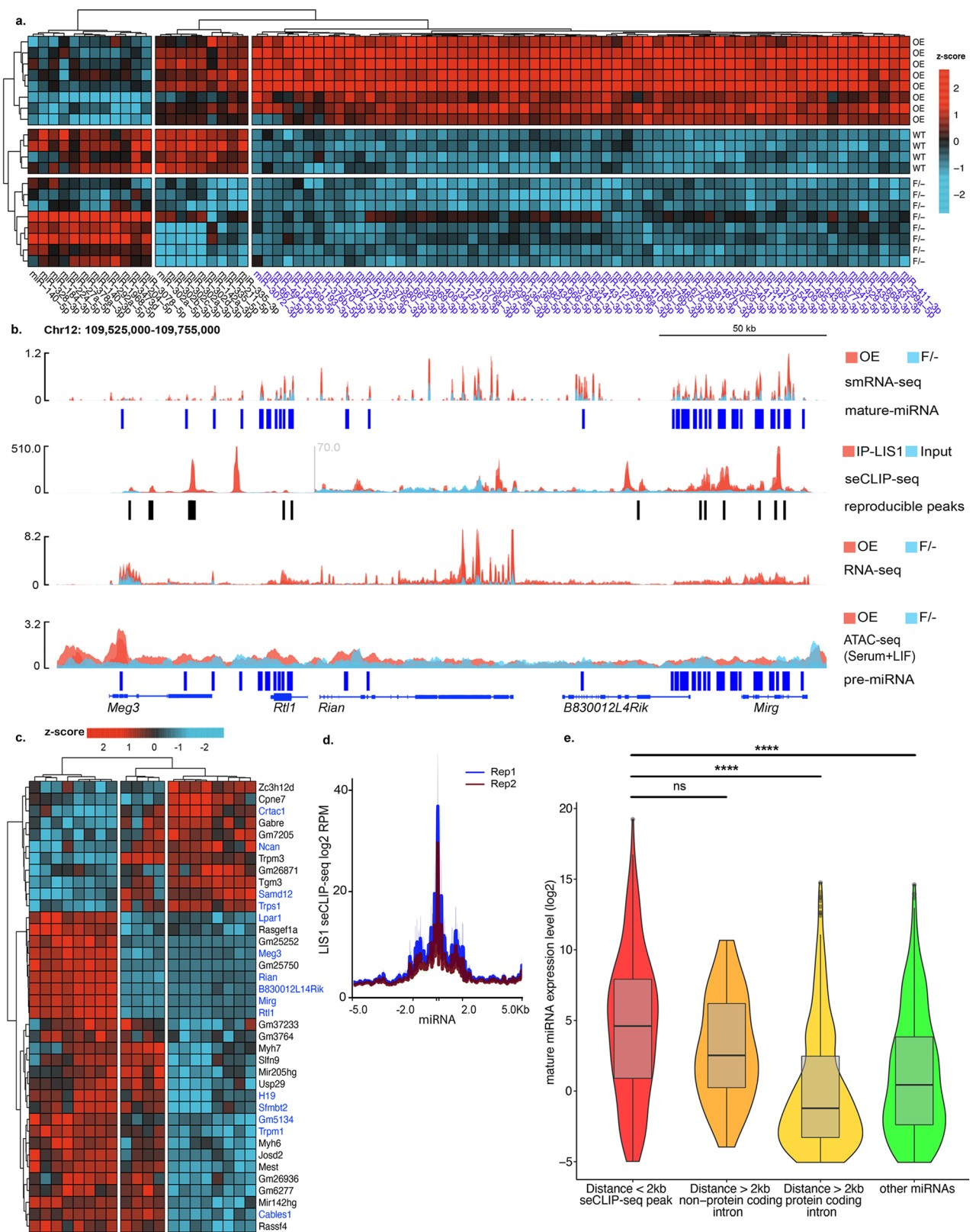

## LIS1 affects the expression of miRs without the Argonaute complex

To further interrogate the interactions between LIS1/AGO2 and the microRNA pathway, we generated an additional cell line where LIS1 is overexpressed (LIS1 OE) in the context of cells lacking Argonaute proteins (LIS1 OE AGO1-4 KO) with the possibility to induce AGO2

expression (LIS1 OE Dox AGO2) (Fig. 5a, b). The AGO1-4 KO line exhibited a paucity of miRs as previously described[54], yet LIS1 OE was sufficient to slightly but significantly increase the expression of a subset of miRs (Fig. 5c, Supplementary Data 11, statistics in Supplementary Data 6d). An additional small subset of miRs whose expression was slightly but significantly increased following LIS1 OE was

**Fig. 4 | Increased LIS1 expression drives the expression of miRs. a** A heatmap of the top 85 DE mature miRs in the comparison between LIS1 overexpressing line (OE) and *Lis1* F/- (F/-) mESCs with Wild-type (WT) is shown. The data are shown on a Z-score scale of the variance stabilizing transformation on normalized reads. **b** *Meg3-Mirg* locus. Top to bottom tracks: small RNA-seq signal (OE in red and F/- in blue); seCLIP-seq (IP in red and input in blue); bulk RNA-seq (OE in red and F/- in blue); ATAC-seq (OE in red and F/- in blue). All tracks are normalized. Of note, the *Meg3* promoter region shows significant differential accessibility between the OE and F/- samples. The plot shows the merged track for replicates of each mESCs non-isogenic clone. **c** A heatmap of differentially expressed miR host genes from bulk RNA-seq. The data are shown on a Z-score scale of the variance stabilizing transformation on normalized reads. **d** Metagene plot of LIS1 seCLIP-seq coverage as a function of distance from all pre-miRs in the mouse genome. **e** Expression (log2 DESeq2 baseMean) of miRs as a function of their distance from the closest LIS1 seCLIP-seq peak. Red; LIS1 seCLIP-seq peaks in the distance <2000 bp ($n = 221$), Orange; LIS1 seCLIP-seq peaks with a distance > 2000bp, the peak resides in the introns of noncoding genes ($n = 40$), Yellow; LIS1 seCLIP-seq peaks with a distance > 2000bp, the peaks reside in the introns of protein-coding genes ($n = 502$), Green; all other miRs ($n = 419$). Boxplots show median and lower or upper quartiles; whiskers show inner fences; violin plots show outer fences. Kruskal−Wallis test and a two-sided Dunn's test for multiple comparisons were performed, *p*-values: \*\**p* < 0.01, \*\*\**p* < 0.001, \*\*\*\**p* < 0.0001, ns-non significant (full comparisons are found in Supplementary Data 6c).

noted in the background of AGO2-expressing cells (Fig. 5d). The relative expression of a subset of miRs was examined by qPCR (Fig. 5e, Supplementary Fig. 10a, statistics in Supplementary Data 6d). Among the 85 DE miRs we previously identified, many are known to regulate tissue stiffness and be involved in mechanosensitivity[66–70], including several members of the let-7 family, miRs 221/222-3p, 146-5p, and 151-5p. In addition, we noted the increased expression of all of the members of the miR 302 family, known to be involved in the pluripotency network in mESCs[71]. The expression of these miRs and others significantly increased in the presence of LIS1 OE (Fig. 5e, Supplementary Fig. 10a).

To explore the possible mechanism of the increased expression of miRs in the absence of all Argonaute proteins, we treated the cells with an RNA polymerase II inhibitor (alpha-amanitin) and tested the expression of *Rian* and miR-314 (Supplementary Fig. 10b). Whereas the steady-state levels of *Rian* decreased in both cell lines, as expected, miR-341 decreased in the AGO 1-4 KO cells but was stable in the LIS1 OE line suggesting that in some cases LIS1 may stabilize selected miRs.

The expression of the let-7 family of miRs may be associated with the fact that they are negatively regulated by LIN28a[72], and this protein is part of the LIS1 interactome (Fig. 2 and Supplementary Data 4c). As indicated above, LIS1 plays a role in splicing. Therefore, we examined the differences in splicing efficiency of miR host genes at the transcript and intron levels in all the cell lines compared to the AGO 1-4KO line (Supplementary Fig. 10 c, d and statistics in Supplementary Data 6e). Significant changes were noted in all comparisons, yet the *p* values were more pronounced when the intron levels were examined. The heatmap of bulk RNA-seq depicted that the expression of AGO2 introduced a significant change to gene expression (Fig. 6a, Supplementary Data 12a). However, in the LIS1 OE AGO1-4 KO, we noticed that the expression pattern of a subset of the genes was similar to that in the DoxAGO2, suggesting that LIS1 OE can partially rescue the dysregulation in AGO1-4 KO cells (Fig. 6a). The GO molecular function terms of the DE genes indicated multiple ECM functions and mechanosensitivity (Fig. 6b). Gene Analytics analysis for the same comparison revealed that LIS1 OE affects the expression of pluripotency genes and differentiation markers (Supplementary Data 12b, Fig. 6c). Based on this analysis, pathways with greatest influence are ERK signaling, ECM, Mesenchymal stem cell, and lineage-specific markers. These pathways are involved in lineage specification and stem cell differentiation.

We then examined local splicing variations (LSV) by comparing the different AGO lines with and without LIS1 OE using MAJIQ (Supplementary Fig. 11a, b, Supplementary Data 13). Reactome pathway enrichment analysis for differentially spliced genes revealed signaling by TGF-beta, RHO GTPase cycle, and chromatin organization (Supplementary Fig. 11c). Examples of alternative splicing events are shown for *Meg3* and *Rian* (Supplementary Fig. 11d, e). We then examined the differences in splicing efficiency between AGO1-4 KO and the rest of the lines in relation to the number of LIS1 seCLIP clusters (Fig. 6d, statistics in Supplementary Data 6f). In the case of the two lines that overexpressed LIS1, we noted that the differences in splicing efficiency

increased with the gain in LIS1 seCLIP clusters. The difference in splicing efficiency following the induction of AGO2 expression in the AGO1-4 KO was not affected by the presence of LIS1 RNA-binding clusters. To examine if AGO2 OE can affect splicing in the *Lis1* F/- cells, we generated a tetracycline-inducible AGO2 line (Supplementary Fig. 12a–c). Analysis of the RNA-seq data showed no difference in the cumulative fraction of the splicing efficiency at the transcript level (Supplementary Fig. 12d). Furthermore, no splicing efficiency changes were noted in relation to the number of LIS1 seCLIP-seq clusters (Supplementary Fig. 12e).

The RNAseq and small RNAseq suggested that the different mESC lines on the AGO1-4 KO background likely differ in their physical properties. To test this hypothesis, we subjected mESC colonies to Atomic Force Microscopy (AFM) nano-mechanical measurements (Fig. 6e, f Supplementary Fig. 13, statistics is in Supplementary Data 6f). The elastic modulus of AGO1-4 KO was the lowest; the addition of LIS1 to these cells (AGO1-4 KO LIS1 OE) exhibited the highest value. The changes in the stiffness are correlated with changes in gene expression observed above (Fig. 6a, b). The Dox AGO2 value was higher than that of the AGO1-4 KO. A potential factor that may affect the mechanosensitivity of the ESCs is YAP. We found that LIS1 OE and/or AGO2 introduction significantly increased YAP expression (Fig. 6g, h).

Our data indicate that LIS1 OE can increase the expression of a subset of miRs and affect gene expression and splicing in an AGO2-independent way. Furthermore, LIS1 OE, together with AGO2 or independent of AGO2, can modulate the stiffness of mESC.

## Discussion

LIS1 has been studied intensively for several decades[73], yet here we show a myriad of novel roles for this protein, especially in relation to post-transcriptional regulation. RNA-seq from LIS1 null cells, cultured in a custom-designed media, indicated the involvement of LIS1 in the regulation of gene expression related to many dynein-mediated activities such as mitosis and microtubule-based transport, adding another level of regulation of dynein functions by LIS1. In addition, we also detected pathways related to organophosphate biosynthetic and fatty acid metabolic processes that were captured in our metabolomics analysis. Several of the affected pathways were associated with RNA. LIS1 dosage also affected gene expression in hESC, with a striking absence of E-cadherin in *LIS1*+/− hESCs evident by immunostainings. Loss of E-cadherin can promote epithelial to mesenchymal transition, affecting the stem cells' pluripotency[74]. To further understand the molecular mechanisms involved in these diverse activities, we have undertaken an unbiased approach to compile the LIS1 interactome in the nuclei and cytoplasm of mESCs with different levels of LIS1. The known LIS1 interactome included only 148 proteins, mainly regulating the cytoskeleton. Our experiment revealed that the LIS1 interactome comprises more than a thousand proteins. Many LIS1-interacting proteins are known RBPs, strikingly representing the Argonaute complex. We further dissected the LIS1 interactome to AGO2-dependent and -independent

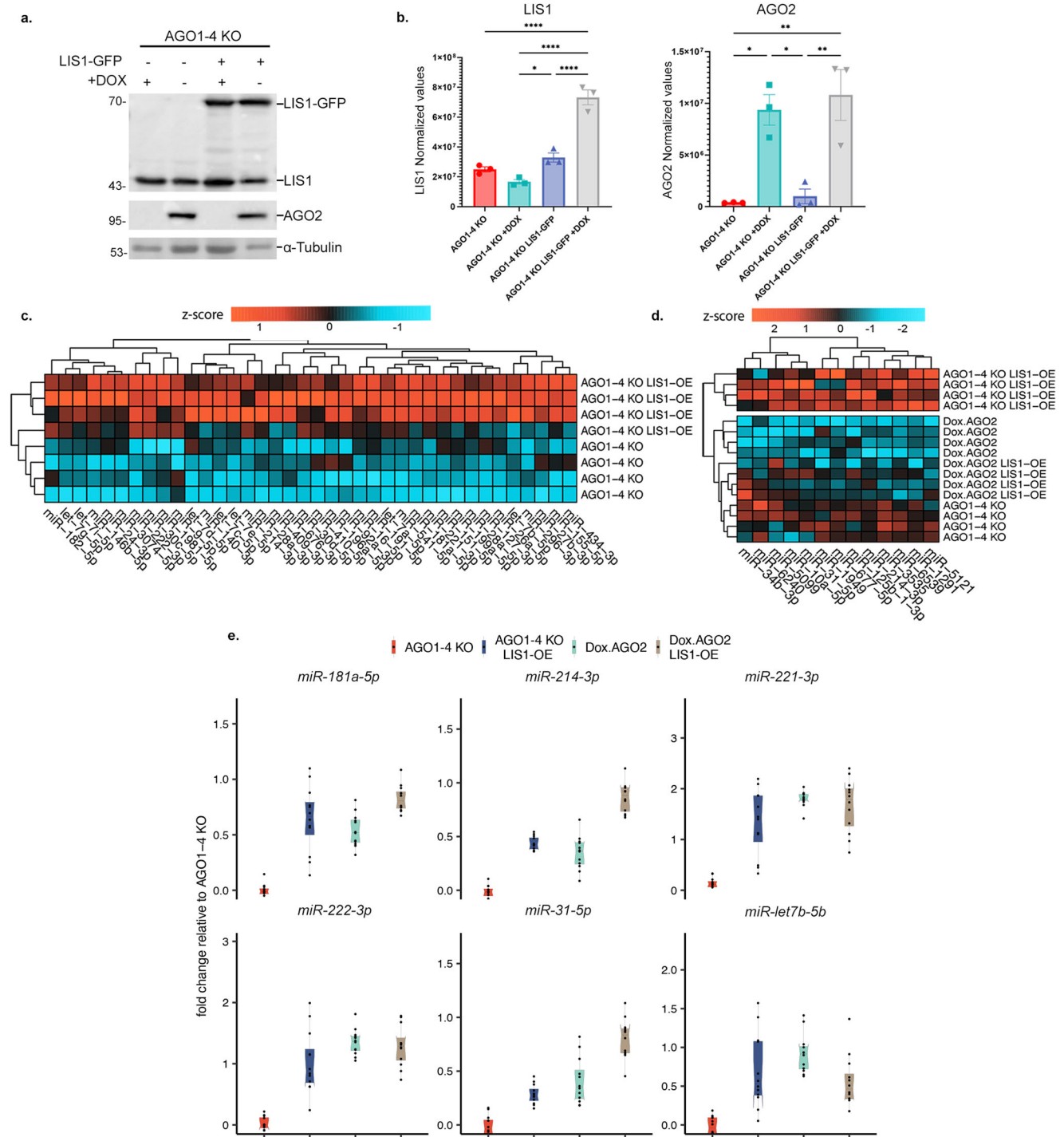

**Fig. 5 | The effects of LIS1 or AGO2 expression on AGO1-4 KO background.**
**a** Western blot of AGO2, LIS1, and α-Tubulin in extracts from AGO1-4 KO, AGO1-4 KO LIS1-GFP (or LIS1 OE), Dox.AGO2, or Dox.AGO2 LIS1 OE. Cells were either treated or not treated with Doxycycline (Dox). **b** Quantification of LIS1 and AGO2 expression levels (*n* = 3, Data are presented as mean values ± SEM.). One-way ANOVA and Tukey's test for multiple comparisons, *p*-values: *\**p* < 0.05, *\*\**p* < 0.01, *\*\*\**p* < 0.001, *\*\*\*\**p* < 0.0001, ns-non significant. **c** A heatmap showing the increase in mature miRs' expression due to LIS1 overexpression (AGO1-4 KO LIS1-OE) compared to AGO1-4 KO. DESeq2 normalized reads with log2 foldchange ≥ 0.5 and adjusted *p*-

values ≤ 0.05 were considered significant. **d** A heatmap showing the increase in mature miR expression due to LIS1 overexpression in AGO1-4 KO LIS1-OE compared to Dox.AGO2, Dox. AGO2 LIS1-OE and AGO1-4 KO from the DESeq2 analysis in (**c**). For (**c**. and **d**.), the data are shown on a Z-score scale of the variance stabilizing transformation on normalized reads. **e** qRT-PCR for a subset of mature miRs involved in regulating ECM and mechanosensitive genes (*n* = 4), boxes show median and lower or upper quartiles; whiskers show inner fences. Kruskal–Wallis test, and a two-sided Dunn's all-paired test for multiple comparisons, were performed; *p*-values are reported in Supplementary Data 6d.

subgroups. The enrichment of RBPs in the LIS1 interactome combined with a previous high throughput study that identified LIS1 as an RNA-binding protein[55] warranted further analysis using seCLIP. An additional literature survey demonstrated that LIS1 was detected as

an RBP in other species[75] including the clawed frog[76], fruit flies[77], and yeast[78], underscoring this evolutionarily conserved property. LIS1 was bound mainly to introns of nascent RNA of protein-coding genes. The presence of an increased number of binding sites within a gene

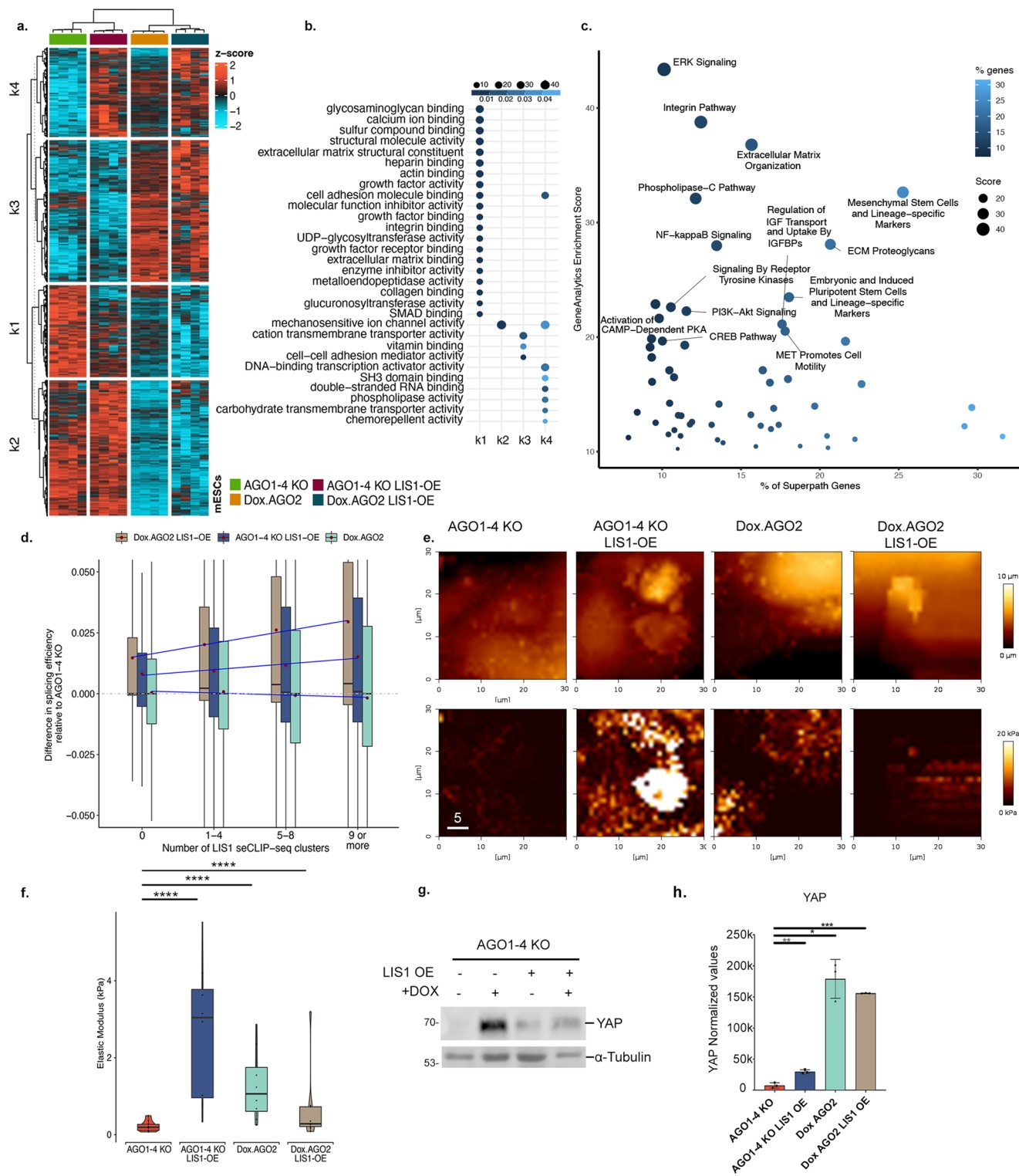

was negatively correlated with its' splicing efficiency, while an increased level of LIS1 was positively correlated with splicing efficiency. The same trend was noted when LIS1 was overexpressed in the background of AGO1-4 KO and Dox AGO2 cell lines. We noted that changes in LIS1 dosage affected splicing in multiple genes, resulting in skipped exons or retained introns, which can affect steady-state RNA levels and protein translation[79].

The association of LIS1 with the Argonaute complex strongly suggests that the expression of miRs might be affected by LIS1 dosage. Furthermore, the LIS1 interactome shares fifty-seven proteins in common with a list of 181 proteins detected in a large-scale biochemical screen to identify a comprehensive list of RBP-miRNA interactions[80] (significant overlap using a hypergeometric test; representation factor: 2.8, $p < 1.980e{-}13$). These RBPs are highly enriched for proteins involved in RNA splicing. Therefore, the effect of LIS1 could be mediated through the RNA-induced silencing complex (RISC) or other LIS1 interacting RBPs. In the context of LIS1 dosage, LIS1 OE significantly increased the expression of multiple miRs, many located in operons. Suboptimal Drosha/DGCR8 substrate miRNAs are enriched in operons, and their proximity facilitates subsequent processing[81].

**Fig. 6 | LIS1 or AGO2 expression impacts cell stiffness in AGO1-4 KO mESCs. a** A heatmap of 3183 genes differentially expressed due to the effect of LIS1 over-expression and doxycycline-induced AGO2 expression in AGO1-4 KO mESCs cultures. The data are shown on a Z-score scale of the variance stabilizing transformation on normalized reads. **b** Analysis of gene set over-representation test for four clusters obtained from the k-means clustering, illustrating the combined effect of LIS1 and AGO2 modulating the expression of genes in top enriched Gene Ontology Molecular Function (GO-MF). **c** GeneAnalytics pathway enrichment analysis for differentially expressed genes between AGO1-KO and AGO1-4 KO LIS1-OE (with overexpression of LIS1) [$n = 1599$]. Top significant pathways identified by GeneAnalytics above an enrichment score of twenty for matched genes in Super-path are shown. **d** Intron level quantification of differences in splicing efficiency for Doxycycline treated AGO1-4 KO lines (Dox.AGO2, green) and LIS1-OE on the background of AGO1-4 KO treated (Dox. AGO2 LIS1-OE, brown) or not treated with DOX (AGO1-4 KO LIS1-OE, blue), relative to AGO1-4 KO and correlated to the number of LIS1 seCLIP-seq clusters. All the comparisons were significant across each genotype in each cluster group; boxplots show median and lower or upper quartiles; whiskers show inner fences. *p*-values for significance are reported in Supplementary Data 6f. **e** Atomic force microscopy images showing height (top row), and corresponding elastic modulus (bottom row) of stem cell coloniesμ (scale bar 5 μm). **f** Box plots showing elastic modulus (in kilopascals (kPa), *y*-axis) for independent and combined effects of constitutive LIS1-OE and Dox.AGO2 in AGO1-4 KO mESCs. Boxes centers show the median, and bounds show lower or upper quartiles of median absolute deviation normalized values for outliers from the measurements of each group; whiskers show inner fences; violin plots show outer fences. A histogram of modulus values for each measurement (6–8 colonies per group) is shown in Supplementary Fig. 13. Kruskal–Wallis and pairwise Games-Howell test for multiple comparisons, *p*-values: \*\*\*\**p* < 0.0001. Statistics in Supplementary Data 6g. **g, h** Western blot analysis of YAP and α-Tubulin expression in AGO1-4 KO lines and LIS1-OE on the background of AGO1-4 KO (LIS1 OE) with and without DOX treatment. **g** Arbitrarily values of YAP expression levels normalized to α-Tubulin expression in indicated mESCs lines (*n* = 3, Data are presented as mean values ± SEM). One-way ANOVA and Tukey's test for multiple comparisons, *p*-values: \**p* < 0.05, \*\**p* < 0.01, \*\*\**p* < 0.001, \*\*\*\**p* < 0.0001.

The protein-coding and non-protein-coding genes in the DE miR loci were also DE. This correlation may be related to the observed changes in chromatin organization evident by ATAC-seq. The expression level of miRs was positively correlated with their distance from LIS1 seCLIP sites. Therefore, our data suggest that LIS1 is involved in the positive regulation of the expression of these miRs, possibly by binding and stabilizing them as indicated by the inhibition of RNA polymerase II experiment (Supplementary Fig. 9b). LIS1 might be involved in the recruitment of additional proteins to the nascent RNA or affecting RNA splicing engaged in the formation of those miRs[82]. However, other mechanisms can contribute to these DE miRs. It has been proposed that impaired export of miRs via exosomes may increase cellular miRs[83]. Considering the known roles of LIS1 in intracellular transport, LIS1 may affect miRs at multiple levels. Most studies focus on the stages of miR biogenesis, but only a few studies investigated the RISC stage. The RISC loading involves the binding of Argonautes to miRs or siRNA duplexes. The passenger strands of the miRs or the siRNA duplexes are degraded, and the mature RISC is guided to the respective mRNA, or Argonautes binds to preformed duplexes of miR/siRNA−mRNA[84,85]. Several proteins are involved in the formation of these AGO-independent duplexes. For example, the RBP AUF1 (HNRNPD) can directly promote the binding of the miRNA let-7b to its target site within the 3'UTR of the POLR2D mRNA[86]. In a systematic eCLIP study examining a set of 126 RBPs, the vast majority (92%) interacted with at least one miR locus[87], suggesting that our understanding of how RBPs regulate miR expression is far from complete.

Our findings indicated that LIS1 OE could increase the stiffness of Argonaute deficient cells. It has been previously demonstrated that AGO2 activity is required for tissue stiffness[66]. Still, it was unknown if the physical properties of the cells could change in an AGO2-independent manner. LIS1 OE in the absence of AGO1-4 resulted in a slight but significant increase in a small set of miRs. Data mining revealed that many of these miRs, including members of the let-7 family, miR-221-3p, 16-5p, 296-3p, and others, are involved in mechanotransduction pathways[67,88–96]. RNA seq data further revealed the DE of genes associated with mechanosensitive pathways. In addition, LIS1 OE in the absence of AGO1-4 resulted in increased YAP expression, which is involved in mechanotransduction pathways. Indeed, in the absence of all Argonaute proteins, LIS1 OE induces a significant elevation in the measured Young's modulus.

Taken together, our studies have demonstrated that changes in LIS1 expression modulate chromatin organization, RNA and small RNA gene expression, and splicing, which results in long-lasting changes in the physical properties of mESCs. We propose that these previously unrecognized, nuclear, RNA- and small-RNA-related LIS1 activities underlie, at least in part, some of the activities that make LIS1 crucial for proper human brain development[1,3,73].

## Methods
### Generation of mESC lines

All animal studies were done following approval of the Institutional Animal Care and Use Committee (IACUC) committee at the Weizmann Institute of Science. The use of experimental animals is in complete accordance with: The Animal Welfare Law (Experiments with animals); The Regulations of the Council for Experiments with Animals; The Weizmann Institute Regulations (SOP); The Guide for the Care and Use of Lab Animals, National Research Council, 8th edition; The Guidelines for the Care and Use of Mammals in Neuroscience and Behavioral Research. All human ESC studies were approved by the Weizmann Institutional Review Board (IRB).

To generate LIS1 mutant mESCs, the *Lis1* F/F (*Lis1^flox/flox*) mice were crossed with mice expressing tamoxifen-inducible Cre with a ubiquitin promoter (*UB-Cre/ERT2*) or with a constitutive maternal Cre expressed under the regulation of the phosphoglycerate kinase promoter (PGK-Cre). *Lis1*+/− without PGK-Cre were crossed with *UB-Cre/ERT2*:*Lis1* F/F to get *UB-Cre/ERT2*:*Lis1* F/- and *Lis1* F/-. LIS1-FLAG-DsRed mice[3] were crossed with PGK-Cre mice to obtain PGK Cre: *Lis1*+/+: LIS1-FLAG-DsRed to obtain ubiquitous LIS1 transgene overexpression. Blastocysts were flushed at embryonic day 3.5 (E3.5) in KSOM medium (Invitrogen, Thermo Fisher Scientific) from naturally mated timed-pregnant mice and cultured for five days in 2i+LIF medium (Supplementary Table 1). All mESCs were routinely maintained on irradiated MEFs in an FBS + LIF medium (Supplementary Table 1). To ensure stable LIS1 over-expression in *UB-Cre/ERT2*:*Lis1* F/- and TT-FHAgo2[97] (received from Drs. Zamudio, Suzuki, and Sharp), the PiggyBac transposase system was used[98]. To generate *UB-Cre/ERT2*:*Lis1-F/-*:LIS1-GFP-OE [LIS1-GFP-OE (F/-)] and TT-FHAgo2:LIS1-GFP-OE (AGO1-4 KO LIS1-OE), we transfected 10 μg of a mix containing the pCAG:*LIS1*-GFP plasmid with a pCAG-PBase plasmid expressing the PiggyBac transposase. We transfected the Xlone-AGO2 *Lis1* flox/- cell line to generate the pB-RFP-Xlone-AGO2 plasmid with a pCAG-PBase plasmid expressing the Pig-gyBac transposase. The pB-RFP-Xlone-AGO2 plasmid was generated by inserting the AGO2 gene into the Xlone-GFP plasmid (AddGene #96930)[99]. The transfection was done using a NEPA21 electroporation system according to the manufacturer's instructions. Following electroporation, cells were passaged once after five days, and GFP-positive cells were selected using fluorescence-activated cell sorting (FACS). To induce the deletion of *Lis1*, cells were treated with 500 nM 4-hydroxytamoxifen (4-OHT) for 72 h in 5i+LIF (Supplementary Table 1) with 100 μM apoptosis inhibitor Z-vad (Sigma-Aldrich®) and 50 μM of necroptosis inhibitor Necrostatin-1 (Sigma-Aldrich®), respectively. The deletion was assessed by genotyping for the floxed and deleted alleles. For mESCs grown in 5i+LIF, the medium was replaced daily. TT-FHAgo2 and TT-FHAgo2 LIS1-GFP-OE clonal cell lines were maintained in FBS + LIF medium and were treated for 72 h

with 0.1 µg/ml doxycycline to induce AGO2 expression. We used the naive pluripotency reporter mESCs line V6.5 deltaPE-Oct4-GFP[100–102], which was grown in 5i+LIF (3–4 passages), and colony morphology was assessed and compared with cells cultured in other media (Supplementary Table 1).

WIBR3 (NIHhESC-10-0079) hESCs and isogenic *LIS1*+/−[10] were grown and maintained on irradiated MEFs in optimal naive NHSM conditions (RSET-Stem Cell Technologies INC, Supplementary Table 1)[101]. Gain of function LIS1-GFP-OE isogenic hESCs were generated with the PiggyBac pCAG:*LIS1*-GFP transfection in WIBR3 hESCs as described above. LIS1 int6/int6 homozygous mutation isogenic hESCs were generated using the CRISPR/Cas9 protocols previously described, with some modifications. Briefly, to introduce cis-splicing mutation, the CRISPR genome-editing method with double Cas9 nickase was used to reduce the probability of off-target effects[103]. The two guide-RNA sequences, sense: ATATTGCTGTTATGTGTTTT and anti-sense: TGGCTACTGAAGAAACATTG, were designed according to http://crispr.mit.edu/ and were cloned into a pX335 vector[103] targeting intron six near the acceptor site of exon 7 of the *LIS1* gene. Cells were transfected using electroporation described above with a repair single-strand oligo cccatggtcaattgatgtttcattgctcttggtggtatattacttcataatatattgctgttaCgtgtttttagGCCATGAtCACAATGTTTCTTCAGTAGCCATCATGCCCAATGGAGATCATATAGTGTCTGCCTCAAGGGATAAAAC encoding the T>C mutation and introducing a silent mutation leading to a new BclI site and the pX335 plasmid with trace amounts of a GFP expression vector. Three days after transfection, the cells were subjected to FACS and plated at a density of 2000 cells per 10 cm plate on irradiated MEFs, allowing for the growth of single-cell-derived colonies. Clone 61 was used in this study. PCR, restriction enzyme digestion, and Sanger DNA sequencing confirmed the mutation. The hESCs were passaged with TrypLE (Thermo Fisher Scientific) every 3–4 days. They were grown on irradiated MEFs for four passages in tHENSM[35] when the media was changed from NHSM (Supplementary Table 1). *LIS1*+/− hESCs lose colony morphology and grow poorly after five passages in tHENSM media.

## Pre-implantation embryo and cell lines immunostaining
E3.5 WT embryos were collected in KSOM media, and after three washes with PBS, embryos were fixed with 4% PFA overnight at 4 °C. The staining procedure was performed in Pyrex spot plates under a Delta vision microscope. ESCs were cultured in multiwell glass-bottom plates (MatTek Life Sciences) coated with Matrigel (Corning Life Sciences) and were fixed with 4% PFA for 20 min at room temperature. Samples were rinsed thrice in PBS and were permeabilized in permeabilization solution (0.1% Triton X-100 in PBS) for 20 min at room temperature. This was followed by incubation in a blocking solution (2% donkey serum or 2% normal goat serum, 0.1% BSA in permeabilization solution). Samples were incubated overnight with primary antibodies in a blocking solution (1:100). The next day, samples were rinsed three times with a blocking solution for 10 min each. This was followed by incubation with secondary antibodies diluted in blocking solution (1:500) for 1 h at room temperature and costained with DAPI (1 µg/ml in PBS) for 5 min. Embryos were moved and allowed to sink in 96-well MatTek plates in PBS. Imaging was done using a spinning disk confocal microscope based on an OLYMPUS IX83 inverted microscope, VisiScope CSU-W1-T1 confocal system (Visitron Systems, Germany), and an sCMOS 4.2 MPixel camera. Imaging was performed using the VisiView software. Imaging of ESCs was carried out using a Dragonfly 200 spinning disk confocal microscope (ANDOR, Oxford instruments). Images were processed using Imaris microscopy image analysis software (Oxford instruments). The following antibodies were used: anti-LIS1 (Sapir T. et al.[26], 338), anti-Nanog (AF2729, R&D), anti-OCT3/4 (C10, Santa Cruz), anti-E-CADHERIN (Abcam, Cambridge, UK), and anti-Pan-LEFTY (Abcam, Cambridge, UK). Image processing and intensity quantifications were done using Imaris© microscopy image

analysis software (Oxford instruments). B-CATENIN membranal localization was measured in ImageJ software. The intensity distributions in nine comparable rectangular regions of interest (ROI) were plotted for each cell line. The coefficient of variance (CV) was calculated by dividing the Standard Deviation (SD) of the measured intensity by the mean intensity of each line; statistical analysis and graphical representation were done using Prism 9 software (GraphPad Software).

## Western blot analysis
Cell lysates were separated by 8-10% SDS–PAGE gels and subsequently electrophoretically transferred from the gel onto a Nitrocellulose membrane. To minimize any unspecific interactions of the antibodies, the membrane was incubated for one h in a blocking solution (5% non-fat milk powder in PBS-T, PBS, and 1% Tween-20) at RT. After a brief wash with PBS-T, the membrane was incubated with primary antibodies overnight at 4 °C and later washed with PBS-T three times for 5 min at RT. Subsequently, the membrane was incubated for 1 h at RT with the secondary antibody at a 1:5000 dilution in the blocking solution (2.5% non-fat milk powder in PBS-T). The membrane was again washed as described above. Antibodies bound to the target protein were detected using the ECL solution (20 ml HCL 8.5 pH, 44 µl p-coumaric acid, and 100 µl luminol). The secondary antibody was used to the primary antibody, Peroxidase AffiniPure Goat Anti-Mouse or Rabbit IgG (H + L) from Jackson (115-035-003 or 111-035-144, respectively).

## Metabolite extraction
mESCs were grown in FBS + LIF medium without Dimethyl 2-oxoglutarate. Cells were trypsinized and centrifuged at 500*g* for 3 min. The medium was removed entirely, and 6×10^6 feeder MEFs depleted ESCs were taken for sample preparation. Extraction and analysis of lipids and polar metabolites were performed as previously described[104,105] with some modifications: samples were mixed with 1 ml of a pre-cooled (−20 °C) homogenous methanol:methyl-tert-butyl-ether (MTBE) 1:3 (v/v) mixture, containing following internal standards: 0.1 µg/ml of Phosphatidylcholine (17:0/17:0) (Avanti), 0.4 µg/ml of Phosphatidylethanolamine (17:0/17:0, 0.15 nmol/ml of Ceramide/Sphingoid Internal Standard Mixture I (Avanti, LM6005), 0.0267 µg/ml d5-TG Internal Standard Mixture I (Avanti, LM6000) and 0.1 µg/ml Palmitic acid-13C (Sigma, 605573). The tubes were vortexed and then sonicated for 30 min in an ice-cold sonication bath (taken for a brief vortex every 10 min). Then, double deionized water (DDW): methanol (3:1, v/v) solution (0.5 ml) containing the following internal standards: C13 and N15 labeled amino acids standard mix (Sigma) were added to the tubes followed by centrifugation. The upper organic phase was transferred into a 2 ml Eppendorf tube. The polar phase was re-extracted with 0.5 ml of MTBE. Both parts of the organic phase were combined, dried in Speedvac, and then stored at −80 °C until analysis. For analysis, the dried lipid extracts were resuspended in 150 µl mobile phase B (see below) and centrifuged again at 17,000*g* at 4 °C for 5 min. The lower polar phase was lyophilized and stored at −80 °C until analysis. Before the injection, the polar phase sample pellets were dissolved using 150 l DDW-methanol (1:1), centrifuged twice (17,000*g*) to remove possible precipitants, and transferred to an HPLC vial were injected into the LC-MS system.

## LC-MS for lipidomics analysis
Lipid extracts were analyzed using a Waters ACQUITY UPLC system coupled to a Vion IMS qTof mass spectrometer (Waters Corp., MA, USA). Chromatographic conditions were as described[104] with small alterations. Briefly, the chromatographic separation was performed on an ACQUITY UPLC BEH C8 column (2.1×100 mm, i.d., 1.7 µm) (Waters Corp., MA, USA). The mobile phase A consisted of DDW: Acetonitrile: Isopropanol 46:38:16 (v/v/v) with 1% 1 M NH₄Ac, 0.1% glacial acetic acid. Mobile phase B composition is DDW: Acetonitrile: Isopropanol 1:69:30

(v/v/v) with 1% 1 M NH$_4$Ac, 0.1% glacial acetic acid. The column was maintained at 40 °C; the flow rate of the mobile phase was 0.4 ml/min, and the run time was 25 min. The linear gradient was as follows: Mobile phase A was run for 1 min at 100%, then reduced to 25% for 11 min, then decreased to 0% for 4 min. Then, mobile phase B was run at 100% for 5.5 min, followed by setting mobile phase A to 100% for 0.5 min. Finally, the column was equilibrated at 100% A for 3 min. MS parameters were as follows: the source and de-solvation temperatures were maintained at 120 °C and 450 °C, respectively. The capillary voltage was set to 3.0 kV and 2 kV for positive and negative ionization mode, respectively; cone voltage was set to 40 V. Nitrogen was used as de-solvation gas and cone gas at a flow rate of 800 L/h and 30 L/h, respectively. The mass spectrometer was operated in full scan HDMS$^E$ resolution mode over 50–2000 Da mass range. A collision energy ramp of 20–80 eV was applied; for the low-energy scan function, 4 eV was used.

### Lipid identification and quantification
LC-MS data were analyzed and processed with UNIFI (Version 1.9.3, Waters Corp., MA, USA). The putative annotation of the lipid species was performed by comparison of accurate mass (below 5 ppm), fragmentation pattern, retention time (RT), and ion mobility (CCS) value to an in-house-generated lipid database. Peak intensities of the identified lipids were normalized to the internal standards and the amount of protein in the cells used for analysis.

### LC-MS polar metabolite analysis
Metabolic profiling of the polar phase was described[11] with minor modifications described below. Briefly, analysis was performed using Acquity I class UPLC System combined with mass spectrometer Q Exactive Plus Orbitrap™ (Thermo Fisher Scientific), operated in a negative ionization mode. The LC separation was done using the SeQuant Zic-pHilic (150 mm × 2.1 mm) with the SeQuant guard column (20 mm × 2.1 mm) (Merck). Mobile Phase B: acetonitrile and Mobile Phase A: 20 mM ammonium carbonate with 0.1% ammonia hydroxide in DDW: acetonitrile (80:20, v/v). The flow rate was kept at 200 μl/min, and the gradient was as follows: 0–2 min 75% of B, 14 min 25% of B, 18 min 25% of B, 19 min 75% of B, for 4 min.

### Polar metabolites data analysis
Data processing was done using TraceFinder (Thermo Fisher Scientific) when detected compounds were identified with accurate mass, retention time, isotope pattern, and fragments and verified using an in-house-generated mass spectra library.

### Immunoprecipitation and mass spectrometry
To identify LIS1 interacting proteins from wild type (WT), Floxed/- (F/-), and LIS1-dsRED OE (OE) mESC, we prepared cytoplasmic and nuclear extracts. mESCs cultured in 15 cm culture dishes were washed with ice-cold PBS. Cells were gently scraped on ice. The cytoplasmic fraction was extracted by incubating in Buffer A (10 mM HEPES pH 7.8, 10 mM KCl, 2 mM MgCl$_2$, 10 mM NaF, 0.05% NP40) supplemented with a protease inhibitor cocktail (Sigma). After centrifugation (for 5 min at 500$g$ at 4 °C), the pellet (the nuclear fraction) was resuspended in twice the volume of high salt buffer (20 mM HEPES pH 7.8, 0.6 M KCl, 2 mM MgCl$_2$, 25% glycerol, 10 mM NaF, protease inhibitor cocktail), incubated on ice for 30 min, and centrifuged for 30 min at 24,000$g$ at 4 °C. For immunoprecipitation (IP), Eppendorf tubes containing A/G protein beads were incubated with monoclonal anti-LIS1 antibodies (20 μl for each sample, 338) in blocking buffer (1xPBS, 0.5% Tween-20, 0.5% BSA) and rotated for 1 h at 25 °C. Next, the nuclear lysate was added and incubated for 6 h at 4 °C in IP buffer (50 mM Tris-HCl pH 7.5, 150 mM NaCl, 1% Triton X-100 and protease inhibitor cocktail) while rotating. Immunoprecipitated proteins were pelleted by centrifugation and washed three times with IP buffer. To identify LIS1 interacting proteins from AGO1-4 KO and Dox.AGO2 mESCs whole-cell lysates (cytoplasm and nuclear extracts) were prepared. Lysates were immunoprecipitated with anti-LIS1 antibodies or IgG antibodies, as described above.

Mass spectrometry was performed as previously described[106] with some modifications. Following the IP with anti-LIS1 antibodies, the precipitated proteins were extracted from the beads using an SDS-sampling buffer. We employed a FASP method to remove SDS from the solubilized protein[107]. Briefly, the SDS samples were applied onto an Amicon filter unit (MWCO = 10 kDa; Ultracel-10 cellulose membrane, Cat.No; UFC201024, Merck Millipore, Billerica, MA, USA) to trap the proteins. After washing the filter units with 8-fold volume (v/v) of Urea buffer (100 mM Tris-Cl (pH8.5), 8 M Urea), the trapped proteins on the filter membrane were alkylated using 10 mM iodoacetamide for 1 hr in the dark. The proteins were digested with Trypsin/ Lys-C (Promega, WI, USA). According to the manufacturer's instructions, demineralization was performed using SPE c-tips (Nikkyo Technos, Tokyo, Japan). The peptides were analyzed by LC/MS using an Orbitrap Fusion mass spectrometer (Thermo Fisher Scientific Inc, MA, USA) coupled to an UltiMate3000 RSLCnano LC system (Dionex Co., Amsterdam, the Netherlands) using a nano HPLC capillary column (Nikkyo Technos Co., Tokyo, Japan) via a nanoelectrospray ion source. Reversed-phase chromatography was performed with a linear gradient (0 min, 5% B; 100 min, 40% B) of solvent A (2% acetonitrile with 0.1% formic acid) and solvent B (95% acetonitrile with 0.1% formic acid) at an estimated flow rate of 300 nl/min.

### Mass spectrometry data analysis
Mass spectrometry raw files were processed using MaxQuant software (version 1.6.2.10)[108] and the Andromeda search engine. Database searches were done against the reference proteome of Mus musculus obtained from UniProtKB in November 2016. The carbamidomethylation of cysteine was set as a fixed modification, and the oxidation of methionine, the phosphorylation of Ser/Thr/Tyr, and N-terminal acetylation were set as variable modifications. Trypsin without proline restriction enzyme option was used, with two allowed miscleavages. False discovery rates (FDRs) for the peptide, protein, and site levels were set to 0.01 with minimal unique+razor peptides number set to 1. The sum of unique+razor peptides across replicates from any fractions with a minimum of two peptides higher than control were considered candidate substrates. Peptide intensity normalization and quantification were performed with the limma algorithm in the DEP Bioconductor package (v.1.16.0)[109]. Mixed imputation was performed on proteins by using a linear model with MAR = zero and MNAR = MinProb. Pairwise differential testing was performed using the test_diff function, and significant proteins were identified using the add_rejection function, setting the $p$-value threshold (alpha) to 0.05.

For the ortholog similarity, *Drosophila melanogaster* and *Mus musculus* ensemble gene pair list was obtained for FlyBase gene identifiers. These were matched with *Mus musculus* homolog-associated gene names using Bioconductor package biomaRt (v2.50.1).

### RNA-seq library preparation
Total RNA was extracted using the miRNeasy Mini kit (Qiagen, Hilden, Germany), and any residual genomic DNA was removed using a DNA-free DNA Removal Kit (Qiagen, Hilden, Germany). RNA concentration and integrity were measured using Nanodrop (Thermo Scientific) and an Agilent Tapestation 4200. Total RNAseq libraries from 1 μg of ribosomal depleted RNA using TruSeq Stranded Total RNA Library Prep (Illumina) or Lexogen SENSE mRNA-Seq Library Prep Kit. These libraries were sequenced on Illumina NextSeq 550 or HiSeq X-Ten to obtain 125 bp or 150 bp paired-end libraries. As previously described, bulk MARS-seq libraries were produced from a modified version of 3' end single-cell RNA-seq[110]. Sequencing libraries were prepared from 50 ng of purified total RNA pooled for biological replicates. The quality

of the final libraries was assessed with qPCR and Agilent TapeStation and was sequenced in the Illumina NextSeq 550 sequencer to obtain a single read of 75 bp.

## RNA-seq analysis

RNA-seq reads were processed using the UTAP pipeline[111]. Briefly, fastq files trimmed with cutadapt and aligned to the GRCm38/mm10 reference genome using STAR (version v2.4.2a), with parameters –alignEndsType EndToEnd, outFilterMismatchNoverLmax 0.05, –twopassMode Basic. Gene read count was performed using the qCount function from the QuasR Bioconductor package (v.1.34)[112] with default parameters. The batch effect of the RNAseq read count was corrected using Combat-seq[113] with a negative binomial regression model on the raw read count data from the sva Bioconductor package (v.3.42). The corrected read count has then used for testing differential expression with the DESeq2[114] Bioconductor package (v.1.34). Group size differences were estimated with the lfcShrink function with -type apeglm[115]. For MARS-seq, samples were analyzed using the UTAP pipeline[111]. Reads were trimmed and aligned to the GRCm38/mm10 and GRCh38/hg38 reference genomes for mouse and human, respectively, as described above. Gene quantification of the most 3′ 1000 bp of each gene was performed using HTSeq-count[116] in union mode while marking UMI duplicates (in-house script and HTSeq-count). Differential expression testing was performed with DESeq2[114] (v.1.34), and pairwise comparison was performed with the lfcShrink function with -type ashr[117]. Genes with log2foldchange $\geq 1$ and $\leq -1$ with padj $\leq 0.05$ and baseMean $\geq 10$ were considered differentially expressed. Clustering was performed with the kmeans function in R. Differentially expressed genes were analyzed using GeneAnalytics[118].

## Splicing efficiency and alternative splicing analysis

Splicing efficiency analysis was performed as described earlier[119]. We used a unique and non-overlapping set of introns for each gene to estimate splicing efficiency at the gene level, confidently supported by the entire RNA-seq dataset. We used only splice-site junction spanning reads for quantification. We defined splicing efficiency as the ratio between exon-exon reads, and all reads (exon-exon plus exon-intron) mapped to junctions. Splicing efficiency values for transcripts and introns were compared using a two-sided t-test. eCLIP clusters (peak regions) were intersected with introns using bedtools[120]. Differential splicing analysis was done using rMATS (v4.0.2)[121] and implementing the MAJIQ-VOILA (v.2.4) pipelines[122,123]. Pairwise rMATS differential alternative splicing events were obtained by options -b1, -b2, -gtf, -t paired --readLength 125 --variable-read-length. For MAJIQ analysis, confounding variations associated with the RNAseq batch were modeled and fitted with MOCCASIN[124]. To quantify local splicing variations (LSVs) and to define splice graphs, the MAJIQ build tool was used, followed by deltapsi and heterogen with default parameters. Results from deltapsi output were further analyzed and parsed with VOILA modulizer and tsv tools. P-values from heterogen output were used for the event-level splice type-specific analysis. Splice graphs were visualized using the VOILA view function, and mis-spliced transcripts between comparisons were defined as significant with ΔPSI (deltaPSI) $\geq 0.2$ and $\leq -0.2$, $p \leq 0.05$. For both algorithms, the gencode.vM25.annotation.gtf transcriptome was used as input.

## Small RNA-seq library preparation

Small RNA libraries were constructed with 1 μg of purified RNA (as described above) using TruSeq Small RNA Library Prep or Lexogen Small RNA-seq Library Prep per the manufacturer's instructions with 14-16 cycles of PCR. Size selection of amplified libraries was made by running on 3% agarose gel followed by gel extraction. The quality of libraries was assessed with Agilent TapeStation before pooling. Truseq small RNA libraries were sequenced to obtain 50 bp single-end reads,

and Lexogen small RNA libraries were obtained to get 75 bp paired-end reads in the Illumina NextSeq 550.

## Small RNA-seq analysis

Adaptor sequences from small RNA-seq libraries were trimmed with cutadapt. Trimmed reads were aligned to the GRCm38/mm10 reference genome using STAR (version v2.4.2a) with parameters --alignEndsType EndToEnd --outFilterMismatchNmax 3 --outFilterMultimapScoreRange 0 --outFilterScoreMinOverLread 0 --outFilterMatchNminOverLread 0 --alignSJDBoverhangMin 1000 --alignIntronMax 1. A custom gtf file was constructed with all mature miRNA annotations from miRbase (v.22) systematically included as distinct transcripts for each parent pri-miRNA gene annotation in gencode.vM25.annotation.gtf to create a txdb object. Primary alignments were counted using the qCount function from the QuasR Bioconductor package (v.1.34) with default parameters. The batch effect was corrected using the ComBat_seq[113] function from the sva Bioconductor package (v.3.42). Normalization and testing for differential expression on corrected read counts were performed with DESeq2[114] on iteratively estimated size factors and mean dispersion estimates with nbinomWaldtest. Raw p-values were adjusted for multiple testing using the Independent Hypothesis Weighting[125] procedure from Bioconductor package IHW. Fold change was estimated manually with the following formula $\log2[(N_x+1)/(N_y+1)]$, where x and y are two conditions, and N is the mean of normalized counts. Only mature miRNA with log2FC $\geq 1$ and $\leq -1$ with padj $\leq 0.05$ and baseMean $\geq 5$ were considered differentially expressed unless specified otherwise. Predictions of miRNA target genes were downloaded from miRDB (v6.0)[126]. Results were filtered to include only miRNA-mRNA pairs with opposing log2FC values in our RNAseq experiments and with miRDB score $\geq 60$.

## ATAC-seq library preparation

ATAC-seq was performed on Lis1 WT, F/-, and OE cells grown in 2i+LIF, FBS + LIF, and FGF+Activin media (Supplementary Table 1) for five passages, as described earlier[127]. Briefly, cells were treated with Trypsin and washed in ice-cold PBS. 50,000 cells were lysed with (10 mM Tris-HCl, pH 7.4, 10 mM NaCl, 3 mM MgCl$_2$, and 0.1% IGEPAL CA-630) ice-cold lysis buffer, and nuclei were spun at 500$g$ for 10 min in a refrigerated centrifuge. Immediately, the pellet was resuspended and incubated in the transposase reaction mix (25 μl 2× TD buffer, 2.5 μl transposase (Illumina), and 22.5 μl nuclease-free water). The reaction was carried out at 37 °C for 30 min. The DNA was purified using the MinElute PCR Purification kit (QIAGEN). After purification, the eluted DNA was amplified for 11-12 cycles with Kappa HiFi Ready-mix (Roche) and custom Nextera PCR primers. The libraries were cleaned and sequenced with a MinELute PCR purification kit.

## ATAC-seq analysis

Nextera adaptors were removed using cutadapt. Bam files were generated by mapping trimmed reads to the genome with Bowtie2 parameters –sensitive local -k 4, and the read pairs separated by more than 2 kb were not considered. Further, post-mapping alignments were filtered by removing reads mapped to chrM, PCR duplicates with PICARD markup, and multimapped reads were removed with samtools -q (MAPQ < 30)[128]. Only short reads ($\leq 130$ bp) corresponding to the nucleosome-free region were considered to identify chromatin-accessible regions. The TOBIAS framework was implemented for subsequent ATAC-seq analysis[129]. ATAC-seq peaks were identified by running MACS2 with parameters --nomodel --shift −100 --extsize 200 --keep-dup all -q 0.01 for each processed bam file. Further, processed bam files for each treatment and condition were merged, and Tn5 transposase insertion bias was corrected with ATACorrect. Footprint scores were generated around merged narrow peak file output from peak calling with ScoreBigwig tools after removing Blacklisted regions

[https://www.encodeproject.org/files/ENCFF547MET/@@download/ ENCFF547MET.bed.gz]. Transcription factor motifs were obtained from HocomocoV11, JASPAR Core 2020, and the JASPAR PBM Homeo collections. Redundant motifs between databases were filtered to one motif for each transcription factor available for the mouse and mouse-human conserved motifs to obtain 645 motifs in JASPAR format. Bias-corrected footprint scores were compared to predict transcription factor binding scores using the BINDetect tool, and a comparison was made between conditions for each treatment and across treatments. Primary alignments were quantified on bias-corrected peak regions using qCount from the QuasR Bioconductor package (v.1.34). Batch correction was performed with Combat_seq and design matrix ~Treatment+Condition:Treatment was used. Differentially accessible regions on corrected primary alignments were obtained with the DESeq2[114] Bioconductor package (v.1.34) using nbinomWaldTest (betaPrior = TRUE). Pairwise contrasts were obtained by creating a model with a group design for condition comparison across each treatment. Raw $p$-values were adjusted for multiple testing using Benjamini and Hochberg procedure[130]. Open chromatin regions with log2foldchange $\geq 1$ and $\leq -1$ with padj $\leq 0.05$ and baseMean $\geq 5$ for condition comparison in each treatment and across treatments were considered differentially accessible ($n = 6984$).

## seCLIP-seq

seCLIP libraries were generated based on the standardized experimental protocol as previously reported[131] with slight modifications. Briefly, V6.5 mESCs were expanded in a 15 cm plate in FBS + LIF media to a density of $30 \times 10^6$ cells. Cells were washed on ice and collected in cold 1X PBS to UV cross-link at 400 mJ/cm$^2$. Sonication was performed for 2 min in a Covaris E220 instrument at intensity 140, burst 200, and duty 5. Lysates were treated with RNase I (1:25 dilution; Ambion) for 3 min at 37 °C. The clarified lysate was transferred to protein G Dynabeads (Invitrogen) conjugated to the monoclonal anti-LIS1 antibodies for two biological replicates. On beads, dephosphorylation was performed with FastAP (ThermoScientific). A 3′ RNA adapter /5′Phos/ rArGrArUrCrGrGrArArGrArGrCrArCrArCrGrUrC/3′SpC3 was ligated to the samples using T4 RNA Ligase (NEB, M0437M) at room temperature for 75 min. Phosphorylation was performed on the beads using PNK (NEB, M0201L) and washed once with wash buffer (20 mM Tris-HCl pH 7.4,10 mM MgCl$_2$,0.2% Tween-20), once with high salt buffer (50 mM Tris-HCl pH 7.4,1 M NaCl, 1 mM EDTA, 1% NP-40, 0.1% SDS, 0.5% sodium deoxycholate) and then twice with wash buffer. Samples were eluted by heating to 70 °C for 10 min in 1X NuPAGE LDS loading buffer and 100 mM DTT at 1200 rpm. Eluates were resolved by denaturing gel electrophoresis and transferred onto a nitrocellulose membrane to determine the migration of Protein-RNA complexes. The relevant region of the separated complexes was cut into small pieces on Whatman paper and transferred to Eppendorf tubes for library preparation. The RNA from the membrane was then isolated by digesting with proteinase K solution [32 units of proteinase K (Invitrogen), 100 mM Tris pH 7.5, 50 mM NaCl, 1 mM EDTA, 0.2% SDS] and incubating at 50 °C for 1 h at 1,200 rpm. RNA was purified by phenol-chloroform extraction followed by ethanol precipitation. The precipitated RNA from the aqueous layer was reverse transcribed using Superscript III reverse transcriptase (Invitrogen) and primer (5′-CAGACGTGTGCTCTTCCGA-3′). A 3′ DNA linker /5′Phos/ NNNNNNNNNNNAGATCGGAAGAGCGTCGTGT/3′SpC3 was ligated onto the cDNA product with T4 RNA ligase. Libraries were amplified with the TruSeq LT adapters, and PCR products were gel-purified using a 3% agarose gel[132].

## seCLIP-seq analysis

seCLIP clusters enriched against size-matched input (SMI) were identified as described previously[133]. Sequenced reads were processed to remove inline barcodes and adaptor sequences. Trimmed reads

mapped to a repeat element database (RepBase v25.06) were removed, and unmapped reads were then mapped to mouse mm10 reference genome assembly using STAR (v2.7.7). Aligned deduplicated reads were merged, and peaks were called using Clipper. Normalized peak files were ranked by entropy score as inputs to IDR to determine reproducible peaks. Reproducible peaks were annotated based on overlap with gencode.vM25.annotation.gtf transcripts. Motif enrichment analysis was performed using the 50 nt sequences flanking the center of seCLIP clusters in both directions. The background sequence sets were generated by HOMER (v4.11)[134] for all possible known motifs. The ngs.plot.r[135] package was used for metaplots. For the eCLIP map of MAJIQ events, coordinates of cassette exons with retained introns and a median ΔPSI difference of 0.2 between conditions were considered. These regions were extended to the intronic region 250 nt upstream of the 3′ splice site plus 50 nt downstream and 50 nt upstream of the 5′ splice site plus the intronic region 250 nt downstream. The seCLIP reads were modeled on the cassette exon extended regions with qProfile functions from the QuasR Bioconductor package (v.1.26.0)[112]. Metaprofiles were normalized to reads per kilobase per million over SMI and averaged over 50 bp for visualization.

For LIS1 seCLIP and miRNA expression joint analysis, intron coordinates for protein-coding genes from gencode.vM25.annotation.gtf were extracted using the extract_pc and extract_introns function from the GencoDymo R package https://doi.org/10.5281/zenodo.3605996. The analysis included all the non-redundant introns and only one assignment from redundant introns. The remaining introns other than protein-coding from gencode.vM25.annotation.gtf were categorized as non-protein-coding introns. All the mature miRNAs from miRbase(v22) showing overlap within 2000 nt distance upstream and downstream from LIS1 seCLIP peaks (<2 kb) were extracted with bed_closest function using valr (v0.6.4)[136] R package. miRNA coordinates overlapping with a distance more than 2000 nt upstream and downstream of LIS1 seCLIP peaks and within introns of protein-coding (>2 kb protein-coding) as well as non-protein-coding (>2 kb non-protein coding) genes were obtained using bed_closest from valr package. Log2 of baseMean values of DESeq2 normalized miRNA counts from LIS1-OE, WT, and F/- for the above three categories were compared with the remaining miRNAs (categorized as other miRNAs). For the LIS1 seCLIP miRNA profile, pre-miRNA coordinates were extended upstream and downstream 2000nt, and the matrix was built using deepTools (v3.5.1)[137]. Defined regions were scaled up to 5000 nt, and binning was done based on the median value.

## AFM experiment and analysis

Atomic force microscopy (AFM) experiments were carried out on TT-FHAgo2 and TT-FHAgo2 LIS1-OE mESCs with and without doxycycline-induced expression of AGO2. Cells were cultured on 5-cm-diameter tissue culture dishes coated with Matrigel (Corning Life Sciences) and grown in Serum+LIF conditions. Cells were allowed to grow for 3 days to reach confluency without colonies touching each other. Fresh media was added before the AFM measurements. AFM imaging and stiffness maps were measured on a Nanowizard III AFM (JPK/Bruker, Berlin, Germany) in QI mode operated with the JPK control software v.6.1.159. In this mode, force-distance curves are collected at each pixel to generate topographic images simultaneously with nano-mechanical data. A BioAC-CI CB3 probe (Nanosensors) was used (nominal spring constant ≈ 0.06 N/m), and sensitivity and spring constant were calibrated before each measurement using the non-contact calibration procedure in the JPK software. The maximum force was chosen to keep penetration depths between 150-500 nm. The elastic modulus maps were calculated using a Hertzian model (JPK data processing software v6.1.86), presuming a Poisson's ratio of 0.5 and a conical tip shape with a 22° half-cone angle. The modulus values were extracted from the modulus maps for the regions in the centers of the colonies, defined by the corresponding height images using Gwyddion software (v2.60)[138].

### miRNA qRT-PCR

The cDNA for qRT experiments was prepared with miScript II RT kit (Qiagen, 218161) using HiFlex buffer. The qRT reactions were performed with miScript SYBR Green PCR kit (Qiagen, 218075). The values of miRNAs were normalized to that of U6 snRNA. The primers are shown in Supplementary Table 2.

### Genotyping

Mice were genotyped at 21 days old by established methods using the primers indicated in Supplementary Table 2.

### Statistics and reproducibility

All bedgraph files were prepared using deepTools (v3.5.1). seCLIP bedgraph were RPM normalized. RNA-seq, small RNA-seq and ATAC-seq bedgraphs were normalized with scaleFactors. For calculating scaleFactors, reciprocals of DESeq2 size factors were estimated using calcNormFactors from edgeR Bioconductor package with RLE method. Track plots were generated from bedgraph files with SparK[139].

All the statistics were performed in base R or using rstatix and statExpressions packages[140] R packages. Gene set over-representation test (p.adj <0.05), and GO term network was done using the Bioconductor package clusterprofiler (v4.2.1)[141]. Schematics related to immunoprecipitation and mass spectrometry were created with Biorender [https://biorender.com/].

All the western blot and immunostaining experiments were performed with minimum $n = 3$ biological replicates. The quantification of the intensities was carried out by ImageJ software. The statistical analysis of measurements was performed using Prism.

### Reporting summary

Further information on research design is available in the Nature Portfolio Reporting Summary linked to this article.

## Data availability

All data, code, and materials used in the analysis are available per request. Stem cell lines and plasmids generated in this study require materials transfer agreements (MTAs). Source data are provided with this paper as a Source Data file. All sequencing data generated have been deposited in the GEO database and are available under accession code GSE198390. The mass spectrometry data have been submitted to the ProteomeXchange database and are available under accession code PXD033150. Source data are provided with this paper.

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

## Acknowledgements

The authors thank Samara Brown, current and previous Reiner and Hanna lab members. Merav Kedmi and Muriel Chemla from the Biological services; Current and ex-staff from the INCPM for help with the library prep and sequencing. Yungui Yang for assistance with sequencing at the Beijing genomics institute. Eran Hornstein and Eli Arama for helpful discussions. Phillip Sharp for the AGO mESCs. Orly Reiner is an incumbent of the Berstein-Mason professorial chair of Neurochemistry and the Head of the M. Judith Ruth Institute for Preclinical Brain Research. Our research has been supported by a research grant from Ethel Lena Levy, a research grant from the Estate of Olga Klein Astrachan, the Weizmann Gladys Monroy and Larry Marks Center for Brain Disorders, the Selsky Memory Research Project, the Advantage Trust, the William and Joan Brodsky Foundation and the Edward F. Anixter Family Foundation, the Helen and Martin Kimmel Institute for Stem Cell Research, the Kekst Center, the Nella and Leon Benoziyo Center for Neurological Diseases, the David and Fela Shapell Family Center for Genetic Disorders Research, the Brenden-Mann Women's Innovation Impact Fund, The Irving B. Harris Fund for New Directions in Brain Research, the Irving Bieber, M.D. and Toby Bieber, M.D. Memorial Research Fund, The Leff Family, Barbara and Roberto Kaminitz, Sergio and Sônia Lozinsky, Debbie Koren, Jack Lowenthal, Lenore Lowenthal, and the Dears Foundation, the Estates of Norman Fidelman, and Hermine Miller, Debbie Koren, Jack, and Lenore Lowenthal, a research grant from the Weizmann SABRA—Yeda-Sela—WRC Program, the Estate of Emile Mimran, and The Maurice and Vivienne Wohl Biology Endowment Canadian Institutes of Health Research (CIHR), the International Development Research Centre (IDRC), the Israel Science Foundation (ISF) and the Azrieli Foundation (2397/18), the ISF grant (545/21), and the United States-Israel Binational Science Foundation (BSF; Grant No. 2017006).

## Author contributions

Conceptualization: A.K., O.R., T.S.; Methodology: A.K., S.M.D., T.O., A.G., T.S., D.T., I.R.G., M.I., A.A., Y.M.G., S.R.C., I.U.; Visualization: A.K., S.M.D., T.O., I.U., A.A., I.R.G., S.R.C.; Funding acquisition: O.R., T.S., K.K., Project administration: T.O., T.S., J.H., Y.M.G., K.K., O.R., Supervision: O.R., T.O., I.U., K.K., Writing—original draft: A.K., O.R., T.O., T.S., Writing—review & editing: A.K., A.G., S.M.D., T.O., T.S., D.T., I.R.G., S.M., M.I., A.A., Y.M.G., S.R.C., J.H., I.U., K.K., O.R.

## Competing interests

The authors declare no competing interests.
