## [Peer Review File · Nature Communications]

LIS1 RNA-binding orchestrates the mechanosensitive properties of embryonic stem cells in AGO2-dependent and independent waysREVIEWER COMMENTS

Reviewer #1 (Remarks to the Author):

In this article Kshrisagar et al. investigate the role of Lis1 in regulating ES cell function. They identify Lis1 as a regulator of AGO2 function in these cells. Their key new finding is defining a novel RNA-binding function for Lis1 in ES cells. The authors carry out extensive experiments, including mass spec and eCLIP to comprehensively determine Lis1 function in both the cytoplasm and nucleus. While these studies are novel and of significant interest in understanding the biology of Lis1, the data, as presented, do not establish the functional role of these different Lis1 activities in regulating ES cell biology and/or cell differentiation and survival. Given that the paper aims to define a novel role of Lis1 in ES cells, it is not clear why the authors chose to primarily focus on LIS1 function in an AGO null context. Are AGO mutations found in conjunction with Lis1 in human patients?

Major Comments

1. The authors identify Lis1 and Oct4/Nanog binding in Fig. 1a. Are these (or any other) stem cell fate determinants also found in the Lis1 nuclear/cytoplasmic IP complex (Fig. 2a/b)?
2. The gene expression changes upon Lis1 over-expression and deficiency in human ESCs indicate changes in genes associated with differentiation and stemness. The authors show that E-cadherin staining is lost with Lis1 deficiency. Does loss in E-cadherin staining also correlate with reduced expression of pluripotency markers such as Oct4 and Nanog, and with an increased propensity to differentiate along specific lineages? Similarly, does Lis1 over-expression lead to increased expression of pluripotency markers?
3. The authors carry out an elegant IP-Mass spec to identify nuclear and cytoplasmic proteins that bind to Lis1. Can the authors corroborate some of their findings, including Ago2 (which they extensively follow up) by IP Westerns in cytoplasmic and nuclear fractions?
4. It is unclear why the authors chose to focus on Ago2. While they do show that the Lis1 interactome shows significant overlap with the Ago2 interactome, it is not clear if no other RNA-binding proteins identified by the mass-spec show a similar overlap in interactomes? It may be beneficial to investigate other proteins with publicly available interactomes (e.g., heat shock proteins, splicing factors) to establish the importance of Ago2 in this context.
5. What are the functional consequences of over-expressing Lis1 in an Ago null background? Does this impair/accelerate the differentiation capacity of the ES cells?
6. The RNA-binding aspect of Lis1 is very exciting. Can the authors speculate on how Lis1 binds to its targets? Does it have canonical or putative RNA-binding domains?
7. At the gene/transcript level, how many of the mRNAs bound by Lis1 (CLIP) show a change in expression by RNA-seq (in Lis1^{-/-} cells)? Do any of these transcripts play a functional role in regulating the Lis1 cellular phenotype (cell death/differentiation)?
8. What is the impact of Lis1 loss (in early passage ES cells) on splicing efficiency as compared to Lis1^{-/+} cells?
9. Can the authors rescue the splicing deficiency in Lis1^{-/+} cells by over-expressing Ago2 and/or other splicing factors that they identify by their proteomics and/or CLIP-seq?

10. Can the Lis1^{-/-} defects and cell death be rescued by expressing miRNAs identified by LIS1 over-expression studies?

Minor Comments

1. The immunofluorescence based Lis1 expression studies in ES cells are of significant interest to the field (Fig. 1a). The authors should quantify Lis1 levels in both cytoplasm and nucleus, as well as the frequency of overlap between Lis1 and Oct4/Nanog.
2. Similarly, it would be of interest to quantify the “truncated” b-catenin expression in Lis1^{+/-} cells (Extended Data Fig. 1d,e). Is this expression further altered in early passage Lis1^{-/-} cells?

Reviewer #2 (Remarks to the Author):

LIS1 (also known as PFAH1B1), famously known as the underlying factor of the neuronal disorder Lissencephaly, has wide ranging functions that are part of normal cell functions, embryonic development and diseases including cancer. Physical interaction between the LIS1 and the cytoplasmic dynein is highly conserved in the eukaryotic organisms from *C. cerevisiae* to human. As the authors mentioned, importance of the LIS1 protein and the structural and mechanistic insights were largely found within the context of its interaction with cytoplasmic dynein and related functions that include mitosis and cell migration.

Kshirsagar et al. set out to discover the novel functions of LIS1 in mouse ESC. The authors took a multidisciplinary approach to analyze RNA-related function of the LIS1. Through the discovery of creating LIS1 KO system in mouse ESC, the authors could first test the impact of LIS1 deletion and rescue by overexpression in mouse ESC gene expression. They first find that both deletion and overexpression of LIS1 result in significant changes in the RNA expression level in mESC and went on to find that LIS1 interacts with numerous RBPs and likely binds the intron of the protein coding genes. Furthermore, they analyzed the miRNA related function of LIS1, especially in relation to that of the AGO proteins in mESC. Taken together, the authors suggest that LIS1 has novel function in posttranscriptional regulation as a RNA binding proteins that bind both coding and non-coding RNAs.

Discovery of LIS1 function in posttranscriptional regulation is novel and as the authors suggest future studies may elucidate its RNA-related function in not only the animal development but also the diseases. The multi-disciplinary approach/analysis taken by the authors provide some of the compelling evidences that suggest importance of direct RNA-protein interaction of LIS1. Nevertheless, it is also important to note that all of the data that authors present are based on the KO/OE of the proteins both of which, as the authors also demonstrate, cause significant changes in the expression level for hundreds of genes, many of which are RBPs. The manuscript could therefore be strengthened by the further validation of direct RNA-protein interaction by LIS1. Related comments and suggestions are detailed below.

Major point

1. Page 5, line 107

Further explanation on the relationship between LIS1 and the metabolites such as fatty acids would be helpful and graphical and statistical analysis of the differentially expressed metabolites could be provided to enhance the understanding of the data.

2. Page 7, line 143 “LIS1 interactome” and Fig. 2a-c.

Based on the result and the figure, it is not completely clear to me how the authors determined LIS1 interactome, which they claim to be 726 proteins. Could the result be presented in similar manner to that of the AGO 1-4KO or Dox.AGO2: IP-LIS1 mass spectrometry analysis result, Figure 1. f-g? More importantly, based on the manuscript including the methods section, it seems that IP experiment was performed without nuclease treatment. Especially since the LIS1 is now considered as a RBP, it would be interesting to perform the same experiment with and without nuclease treatment to distinguish/exclude the RNA dependent interaction from that of the protein-protein interactions, as it is commonly done for the RPB IP-MS studies (Li et al. 2016)

3. Page 9, line 190 “previous high throughput studies identified LIS1 as an RBP”

LIS1 does not harbor any of the known classical RBDs but is one of the WD40 repeat proteins, which was implicated in RNA binding activity (Jin et al. 2016, Castello et al. 2012). It would be more convincing if the authors could further elaborate on the basis of RNA-protein interaction conducted by LIS1 including if the protein is identified in more than one RNA interactome profiling studies, apart from the one the authors cited, conducted in the mouse and human cells or other model organisms (Hentze et al. 2018, Panhale et al. 2019, Na et al. 2021).

4. Page 9, line 203

Could the LIS1-RNA interaction have specificity to the GU-rich sequence? If so further experimental validation or data analysis to show that LIS1's sequence dependent RNA interactions, which is independent of the fact that LIS1 binds the donor splice site, would serve as a compelling evidence to indicate direct RNA binding activity of LIS1.

5. Figure 5h

It isn't clear to me why the authors chose not to perform similar comparison with the wild type and/or LIS1 KO or OE cells. The authors suggested that AGO 1-4 KO “likely” alter the physical properties of the cells and does not provide the data supporting that assumption. It is also puzzling that Dox.AGO2 & LIS1-OE resulted in slightly lowered elastic modulus, in other words no definitive conclusion can be drawn from this result, other than the fact that LIS1 OE can affect the stiffness of mESCs in one way or another, which is rather expected based on the known function of LIS1 protein.

Minor point

1. Figure 1d. and other related figure.

Color code and the size of dots are hard to distinguish and the meaning of number need to be explained on the figure legend.

2. Page 4, line 87, Page 5, line 97 "(Fig. 1B)" – Inconsistent letter case.

References

Jin W, Wang Y, Liu CP, et al. Structural basis for snRNA recognition by the double-WD40 repeat domain of Gemin5. *Genes Dev.* 2016;30(21):2391-2403. doi:10.1101/gad.291377.116

Castello A, Fischer B, Eichelbaum K, et al. Insights into RNA biology from an atlas of mammalian mRNA-binding proteins. *Cell.* 2012;149(6):1393-1406. doi:10.1016/j.cell.2012.04.031

Yang Li, Mahlon Collins et al. Immunoprecipitation and mass spectrometry defines an extensive RBM45 protein–protein interaction network, *Brain Research*, Volume 1647, 2016, Pages 79-93, ISSN 0006-8993, <https://doi.org/10.1016/j.brainres.2016.02.047>.

Panhale, A., Richter, F.M., Ramírez, F. et al. CAPRI enables comparison of evolutionarily conserved RNA interacting regions. *Nat Commun* 10, 2682 (2019). <https://doi.org/10.1038/s41467-019-10585-3>

Na Y, Kim H, Choi Y, et al. FAX-RIC enables robust profiling of dynamic RNP complex formation in multicellular organisms in vivo. *Nucleic Acids Res.* 2021;49(5):e28. doi:10.1093/nar/gkaa1194

Hentze MW, Castello A, Schwarzl T, Preiss T. A brave new world of RNA-binding proteins. *Nat Rev Mol Cell Biol.* 2018;19(5):327-341. doi:10.1038/nrm.2017.130

Reviewer #3 (Remarks to the Author):

In the manuscript titled "LIS1 RNA-binding orchestrates the mechanosensitive properties of embryonic stem cells in AGO2-dependent and independent ways", the authors used multiple high-throughput methods to characterize how different doses of LIS1 impact post-transcriptional regulation in embryonic stem cells (ESCs). The authors uncovered several interesting properties of (nuclear) LIS1 such as its interaction with AGO2 and many RNA-binding proteins (RBPs) and its possible regulatory role in RNA splicing and microRNA (miRNA) expression. With the potential caveat of LIS1 overexpression, it would have been an impactful work if the authors could have provided a clear example on how, at the physiological level, LIS1 participates post-transcriptional regulation in ESCs with mechanistic information. However, the manuscript is mostly descriptive and in some part lacks a coherent narrative regarding the biological functions of LIS1 in post-transcriptional regulation. Overall, the manuscript in its current form unfortunately is not a strong candidate to be considered for publication in *Nature Communications*.

Nevertheless, the current work does raise several interesting questions: What are the (strongest) effects of LIS1 on gene expression (including mRNAs and small RNAs)? How does it affect gene expression mechanistically (direct/indirect)? How does this previously unknown role in gene expression fit into the known phenotype from LIS1 mutations? Those are not easy questions but if the authors could provide

insights into some of the aforementioned questions, possibly through additional biological investigations guided by the existing multi-modality data, it will surely elevate the impact from the current work.

#Main comments:

1) p.4~6, Paragraph: Dosage sensitive effects of LIS1 in pluripotent networks: What's the expression level of LIS1 OE in both mouse and human ESC? From the Methods section, CAG promoter, a strong promoter, is used to drive LIS1-GFP expression. I understand that the authors set out to investigate the LIS1 dosage effects and rescue the lethal effects from LIS1 deletion. However, LIS1-OE mESC are used to investigate effects on gene expression in many of the following sections, and supra-physiologic level of LIS-GFP expression can have effects on gene expression that are not present at the endogenous level. The authors need to perform western blot to directly compare LIS1/LIS1-GFP expression and estimate the dosage of LIS1 across all the cell-lines used. This will provide an important reference to the discovered effects on gene expression.

2) p.4~6, Paragraph: Dosage sensitive effects of LIS1 in pluripotent networks: Is it possible to provide a comparison of the DE genes between mESC and hESC (with comparable dosage effects)?

3) p.9, Extended Data Fig. 3b/c:

-The comparison is to genome, what about unbound (but expressed) genes (i.e., non-target)? Is it still significant?

-I know the information is provided in the supplemental table, but it will help to have the p-value on the plot as well.

4) p.9~10, Paragraph: LIS1 is an RNA-binding protein: Are the mESC RNA-seq data presented in this paragraph (OE, WT, F/-) new or are they part of the RNA-seq data described in Fig. 1c? Please clarify in the text. If they are not new, please use the same exact designation (including 4-OHT usage) from Fig. 1c.

5) p.9, "...We detected a negative correlation between the number of LIS1 seCLIP sites and splicing efficiency in wild-type RNAseq data". Were Lefty2 and Cdh1 splicing efficiency affected? If yes, please add the RNA-seq data to Fig. 3d/e to show the splicing phenotype. If no, would it be possible to provide other examples to show that LIS1 intronic binding affects splicing efficiency of the target genes?

6) p.10, "...Retained intron events were plotted, showing more events occurring in the F/-". Does this mean there are more intron retention events in F/- compared with OE? If LIS1 binding is negatively correlated with splicing efficiency, isn't the expectation to be MORE intron retention in OE? Please clarify and explain.

7) p.9~10, Paragraph: LIS1 is an RNA-binding protein:

-The authors need to fully integrate of LIS1 seCLIP data with RNA-seq data to identify how direct LIS1 binding affects splicing in candidate genes. Does direct LIS1 binding block or promote splicing? Just like other RBPs, LIS1 can do both in different genes but the authors need to parse their data further to find

out.

-One caveat about existing data: it is a good choice to perform LIS1-seCLIP in wildtype mESC with physiological level LIS1 expression. However, overexpression may result in additional binding not seen at physiologic conditions. If LIS1 is expressed a lot higher under LIS1-OE condition, the observed splicing phenotype from LIS1-OE can be caused by binding not captured in the existing LIS1-seCLIP data.

8) p.9~10, Paragraph: LIS1 is an RNA-binding protein: Is there any splicing difference detected between WT and F/-? The information should also be provided even if there are very few changes, as the disease phenotype, lissencephaly, is caused by LIS1 loss-of-function rather than overexpression.

9) p. 11~12, Paragraph: LIS1 affects the expression of miRs: Same as 8). What's the rationale focusing on the DE list from LIS1 overexpression? In Fig. 4a, it looks like there is a group of miRs express lower in F/- compared to WT and OE (middle). The authors need to comment on this group of miRs as well.

10) p.12~13, Paragraph: LIS1 affects the expression of miRs without the Argonaute complex: It is very intriguing that LIS1 can upregulate a subset of miRs in the absence of Ago2. miR expression is complex and involves multiple steps. However, it will elevate the impact of the manuscript if the authors can provide any example as to how LIS1 affects miR expression in the absence of Ago2 mechanistically.

#Minor comments:

1) Fig. 3h legend: Please define upregulation / downregulation: which genotype relative to which genotype?

2) Extended Data Fig. 3a: What antibody was used for this western blot? The same anti-LIS1 used for IP? If the labels are correct, the western blot didn't quite show a clean IP with LIS1 only. What are the additional bands in the IP lanes? Since LIS1 does interact with many other RBPs, how do you make sure there is only LIS1 in your IP?

3) Extended Data Fig. 4d/e: Please provide more explanation for the various panels (especially panels at the bottom row): It's not clear what are the difference in splicing between F/- and OE from the busy plots.

NCOMMS-22-12510-T REVIEWER COMMENTS

We thank the anonymous reviewers for their helpful comments and suggestions which markedly improved our manuscript.

Reviewer #1 (Remarks to the Author):

In this article Kshrisagar et al. investigate the role of Lis1 in regulating ES cell function. They identify Lis1 as a regulator of AGO2 function in these cells. Their key new finding is defining a novel RNA-binding function for Lis1 in ES cells. The authors carry out extensive experiments, including mass spec and eCLIP to comprehensively determine Lis1 function in both the cytoplasm and nucleus. While these studies are novel and of significant interest in understanding the biology of Lis1, the data, as presented, do not establish the functional role of these different Lis1 activities in regulating ES cell biology and/or cell differentiation and survival. Given that the paper aims to define a novel role of Lis1 in ES cells, it is not clear why the authors chose to primarily focus on LIS1 function in an AGO null context. Are AGO mutations found in conjunction with Lis1 in human patients?

We modified the text:

To assess the conservation of the identified LIS1-interactome, we compared our results with a previously published large-scale *Drosophila* interactome ¹. A significant number of the translated *Drosophila* orthologs overlapped with the LIS1-interacting proteins in mESCs (Supplementary Fig. 4G). These included members of the AGO clade, heat-shock, chaperone proteins, and several splicing factors, suggesting that these interactions are evolutionarily conserved (Fig. 2D). We further detected a significant overlap between the nuclear interactome of LIS1 and AGO2, and indicated common and distinct LIS1 and/or AGO2 processes ² (Supplementary Fig. 4 H, I). For example, we detected “transcription repression” and “extracellular exosome” among the common GO terms. Following this significant overlap and the findings that mutations in *AGO2* result in intellectual disability and developmental delay ³, we reasoned that at least part of LIS1 functions might be mediated through its interaction with AGO2.

Major Comments

1. The authors identify Lis1 and Oct4/Nanog binding in Fig. 1a.

We did not claim to identify Oct4/Nanog binding in Fig. 1a. This panel indicates the partial colocalization of these proteins in the early blastocyst stage. It is possible that these proteins can be found in a complex, but the results supporting this claim were not consistent across different embryonic stem cell lines. In the case of Oct4 (encoded by POU domain, class 5, transcription factor 1, Pou5f1) several peptides were noted in the MS data of the AGO lines (Supplementary Table 5B). Nevertheless, we do not wish to pursue this avenue and focused on the LIS1-AGO2 interaction.

Are these (or any other) **stem cell fate determinants** also found in the Lis1 nuclear/cytoplasmic IP complex (Fig. 2a/b)?

Our mass spec data identified several stem cell fate determinants, which we added in Supplementary Table 5C. These proteins include: H2BC9, PABPN1, SRRM1, RNMT, SRRT, SRSF1, NACC1, MOV10, DDX21, UBTF, CSNK2A1, TAF15, U2AF1, U2AF2, PSMA5, PSMB4, SRSF2, SRSF3, SRSF5, SRSF6, SRSF7, DNMT3L, RPS27A, YBX1, TJP1, SRSF9, YWHAE, YWHAG, YWHAH, YWHAZ, SSB, PSMB5, SMAD2, PSMC2, PSMC6, PSMD14, PSMD4, PSMD6, PSME1, PSME2, DEK, ALYREF, NUDT21, MYBBP1A, POU5F1, SOX2, WDR5, RPA1, RPA2, RRM2, LGALS3, CNOT9, MYO1C, PIN1, NR2E3, SF3B1, SNRPB, SNRPD3, SNRPE, SNRPF, SNRPG, SIRT1, SKP1, AGO1, AGO2, NOP2, NCBP1, SMARCA5, PRMT1, MAGOH, CDKN1B, TNRC6A, TNRC6B, TNRC6C, FKBP5, NPM1, H2AZ2, CPSF6, HSPD1, and RYBP.

2. The gene expression changes upon Lis1 over-expression and deficiency in human ESCs indicate changes in genes associated with differentiation and stemness. The authors show that E-cadherin staining is lost with Lis1 deficiency. Does loss in E-cadherin staining also correlate with reduced expression of **pluripotency markers** such as Oct4 and Nanog and with an increased propensity to differentiate along specific lineages? Similarly, does Lis1 over-expression lead to increased expression of pluripotency markers?

We have added Gene analytics pathway enrichment analysis in Supplementary Table 4B for RNA-seq data and observed changes related to pluripotency and differentiation pathways such as Nanog in mammalian pluripotency, Oct4 in mammalian pluripotency, Neural Stem Cell and lineage-specific markers, Neural crest cells differentiation, Mesodermal commitment pathway, Endoderm differentiation, etc. In addition, we examined the expression of OCT4 and NANOG in human cells and in mouse cells.

The text was modified (two different sections):

We derived additional F^{-/-}, WT, and LIS1-DsRED overexpression (OE) mESC lines from blastocysts and immunostained for OCT4. We observed that LIS1 OE reduced the presence of OCT4 in the nucleus (Supplementary Fig. 1D-E).

Gene Analytics pathway analysis revealed genes involved in the extracellular matrix organization, pluripotency, and others (Supplementary Table 4B). A subset of the differentially expressed genes was common to mouse and humans (Supplementary Table 4C). We confirmed the differential expression of a few of the genes by immunostaining. The pluripotent transcription factors OCT4 and NANOG levels were increased in cells with higher LIS1 dosage, with the highest in LIS1-OE hESC (Supplementary Fig. 3B,C, Statistics in Supplementary Table 7A). With its central role in embryonic stem cell pluripotency, epithelial to mesenchymal transition (EMT), and mechanotransduction, E-cadherin was of particular interest^{4,5,6}. *LIS1*^{+/-} cells did not express E-cadherin, though the control and the LIS1-OE cells expressed it at high levels (Fig. 1H, Supplementary Fig. 3D).

3. The authors carry out an elegant IP-Mass spec to identify nuclear and cytoplasmic proteins that bind to Lis1. Can the authors corroborate some of their findings, including Ago2 (which they extensively follow up) by IP Westerns in cytoplasmic and nuclear fractions?

We performed IP experiments followed by western blot and observed a strong reciprocal interaction between LIS1 and AGO2, as shown in Figure 2E.

The added text:

We confirmed the interaction between LIS1 and AGO2 by co-immunoprecipitation followed by western blot in the nuclear and cytoplasmic fractions (Fig. 2E).

4. It needs to be clear why the authors chose to focus on Ago2. While they do show that the Lis1 interactome shows significant overlap with the Ago2 interactome, it is not clear if **no other RNA-binding proteins identified by the mass-spec show a similar overlap in interactomes?** It may be beneficial to investigate other proteins with publicly available interactomes (e.g., heat shock proteins, splicing factors) to establish the importance of Ago2 in this context.

We show a strong interaction with AGO complexes in both nuclear and cytoplasmic fractions. We also observe that the LIS1 interaction with AGO2 is not disrupted following benzonase nuclear treatment suggesting that the interaction is independent of RNA or DNA (Figure 2E, F).

Data from BIOGRID for the interactomes of 31 LIS1 interacting RNA-binding proteins (that appear in Figure 2C) and 16 LIS1 interacting heat shock proteins (a total of 47 proteins) were retrieved. We calculated the overlap between the LIS1 interactome (including a combined list of our data set and the existing data set downloaded from BIOGRID) and the interactome of each protein mentioned above. We performed a hypergeometric test and corrected for multiple comparisons (FDR). There was a significant overlap between the LIS1 interactome and all of the tested proteins. The most prominent overlaps were noted with the RNA binding motif protein 39 (RBM39) and elongation factor Tu GTP binding domain containing 2 (EFTUD2) interactomes.

Nevertheless, for these proteins, there is no information regarding the distribution of their interacting proteins in the nucleus and the cytoplasm.

We have added supplementary Tables 5D- 5E for the most significant interacting proteins from BioGRID database.

The text was modified:

Nevertheless, we note significant overlaps between the LIS1 interactome, and the interactomes of thirty-one RBPs and sixteen LIS1 interacting heat shock proteins (depicted in Fig. 2C and Supplementary Table 5D-E).

5. What are the functional consequences of over-expressing Lis1 in an Ago null background? Does this impair/accelerate the differentiation capacity of the ES cells?

We have performed a western blot experiment and observed changes in the levels of YAP and Nanog, as shown in Figure 6 F-G.

We performed gene analytics enrichment in the supplementary Table 13 B for the RNA-seq data comparing AGO1-4 KO and AGO1-4 KO with overexpression of LIS1. We found that LIS1 OE affects the expression of pluripotency genes and differentiation markers. Based on this analysis, we see that the top pathways affecting are ERK signaling, ECM, Mesenchymal stem cell and lineage-specific markers, etc. These pathways are involved in lineage specification and stem cell differentiation.

The text was modified:

Gene Analytics analysis for the same comparison revealed that LIS1 OE affects the expression of pluripotency genes and differentiation markers (Supplementary Table 13B). Based on this analysis, we see that the top pathways affecting are ERK signaling, ECM, Mesenchymal stem cell and lineage-specific markers. These pathways are involved in lineage specification and stem cell differentiation.

6. The RNA-binding aspect of Lis1 is very exciting. Can the authors speculate on how Lis1 binds to its targets? Does it have canonical or putative RNA-binding domains?

We added to the text:

Furthermore, a previous high-throughput study identified LIS1 as an RBP⁷. The interaction of LIS1 with RNA maps to amino acids 72-88 (LNEAKEEFTSGGPLGQK), contained within the N-terminal domain of LIS1 and is not part of the WD repeats.

7. At the gene/transcript level, how many of the mRNAs bound by Lis1 (CLIP) show a change in expression by RNA-seq (in Lis1^{-/-} cells)?

We want to emphasize that the Lis1 ^{-/-} cells culture condition differs from WT's (V6.5), and the RNA binding could be context specific. Still, we see the overlap of 1627 LIS1-bound differentially expressed genes for LIS1^{-/-} cells from eCLIP data (Supplementary Table 8 B.)

The text was modified:

Part of the LIS1-bound genes was also DE (Supplementary Fig. 6A-F, Supplementary Fig. 7A, Supplementary Table 8B).

Do any of these transcripts play a functional role in regulating the Lis1 cellular phenotype (cell death/differentiation)?

We have added Supplementary Table 8C showing Gene analytics pathway enrichment results. We see that the top pathways affecting include DNA Damage, Apoptotic Pathways in Synovial Fibroblasts, Gene Expression (Transcription), ERK Signaling, Nervous system development, Wnt / Hedgehog / Notch, Mesodermal Commitment Pathway, MiRNA Regulation of DNA Damage Response, Transcriptional Regulation of Pluripotent Stem Cells.

We modified the text:

Gene analytics pathway enrichment analysis showed that the top affected pathways included DNA Damage, Apoptotic Pathways in Synovial Fibroblasts, Gene Expression (Transcription), ERK Signaling, Nervous system development, Wnt / Hedgehog / Notch, Mesodermal Commitment Pathway, MiRNA Regulation of DNA Damage Response, and Transcriptional Regulation of Pluripotent Stem Cells (Supplementary Table 8C).

8. What is the impact of *Lis1* loss (in early passage ES cells) on splicing efficiency as compared to *Lis1*^{-/+} cells?

Lis1^{-/-} cells do not survive for more than 3-4 passages. Differential expression analysis was performed on the MARS-seq data generated from these samples, not the total RNAseq data. The amount of RNA we can retrieve from cells without LIS1 is minimal, and splicing information is unavailable in the MARS-seq.

The text was modified:

The MARS-seq data generated from the *Lis1*^{-/-} cells (Fig. 1C) is incompatible with the splicing analysis. Nevertheless, we noted differential expression of a large cohort of RNA binding proteins, including splicing factors such as *RBFOX2*, *SNRNPA1*, *SNRNPD1*, *HNRNPC*, *HNRNPH3*, and *HNRNPM*, indicative of global changes in splicing.

9. Can the authors rescue the splicing deficiency in *Lis1*^{-/+} cells by **over-expressing Ago2** and/or other splicing factors that they identify by their proteomics and/or CLIP-seq?

We generated a new embryonic stem cell line with *Lis1* F/- ERT2 background where we cloned AGO2 in tet-inducible (Xlone) piggybac plasmid for over-expression of Ago2, performed total RNAseq and analyzed splicing efficiency.

We do not observe any significant difference in splicing efficiency at the transcript level and by the number of LIS1 seCLIP clusters with increased expression of AGO2, as shown in Supplementary Figure 11.

The text was modified:

To examine if AGO2 OE can affect splicing in the *Lis1* F/- cells, we generated a tetracycline-inducible AGO2 line (Supplementary Fig. 11A-C). Analysis of the RNA-seq data showed no difference in the cumulative fraction of the splicing efficiency at the transcript level (Supplementary Fig. 11D). Furthermore, no splicing efficiency changes were noted in relation to the number of LIS1 seCLIP-seq clusters (Supplementary Fig. 11E).

10. Can the *Lis1*^{-/-} defects and cell death be rescued by expressing miRNAs identified by LIS1 over-expression studies?

This is an excellent idea involving a wide range of screens. However, that is beyond this manuscript's scope (We estimate this experiment can take at least 2 years).

Minor Comments

1. The immunofluorescence based *Lis1* expression studies in ES cells are of significant interest to the field (Fig. 1a). The authors should quantify *Lis1* levels in both cytoplasm and nucleus, as well as the frequency of overlap between *Lis1* and Oct4/Nanog.

We performed immunostainings for Oct4 in mouse ES cells with LIS1 different dosages, as shown in Supplementary Figure 1 D, E.

The text was modified:

We confirmed the interaction between LIS1 and AGO2 by co-immunoprecipitation followed by western blot in the nuclear and cytoplasmic fractions.

2. Similarly, it would be of interest to quantify the "truncated" b-catenin expression in *Lis1*^{+/-} cells (Extended Data Fig. 1d,e). Is this expression further altered in early passage *Lis1*^{-/-} cells?

The term truncated is not relevant here. We see a difference in the localization of B-catenin. We did not perform immunostainings in the mouse *Lis1*^{-/-} cells. We investigated the status of b-catenin in hESCs with different LIS1 levels. These results are shown in Supplementary Fig. 3E-H.

The text was modified:

We examined the localization and expression of β -Catenin, the primary downstream target of the Wnt pathway, in hESCs and found the most striking difference in *LIS1*^{+/-} colonies, where the protein appeared mislocalized (Supplementary Fig. 3E-H).

Reviewer #2 (Remarks to the Author):

LIS1 (also known as PFAH1B1), famously known as the underlying factor of the neuronal disorder Lissencephaly, has wide ranging functions that are part of normal cell functions, embryonic development and diseases including cancer. Physical interaction between the LIS1 and the cytoplasmic dynein is highly conserved in the eukaryotic organisms from *C. cerevisiae* to human. As the authors mentioned, importance of the LIS1 protein and the structural and mechanistic insights were largely found within the context of its interaction with cytoplasmic dynein and related functions that include mitosis and cell migration.

Kshirsagar et al. set out to discover the novel functions of LIS1 in mouse ESC. The authors took a multidisciplinary approach to analyze RNA-related function of the LIS1. Through the discovery

of creating LIS1 KO system in mouse ESC, the authors could first test the impact of LIS1 deletion and rescue by overexpression in mouse ESC gene expression. They first find that both deletion and overexpression of LIS1 result in significant changes in the RNA expression level in mESC and went on to find that LIS1 interacts with numerous RBPs and likely binds the intron of the protein coding genes. Furthermore, they analyzed the miRNA related function of LIS1, especially in relation to that of the AGO proteins in mESC. Taken together, the authors suggest that LIS1 has novel function in posttranscriptional regulation as a RNA binding proteins that bind both coding and non-coding RNAs.

Discovery of LIS1 function in posttranscriptional regulation is novel and as the authors suggest future studies may elucidate its RNA-related function in not only the animal development but also the diseases. The multi-disciplinary approach/analysis taken by the authors provide some of the compelling evidences that suggest importance of direct RNA-protein interaction of LIS1. Nevertheless, it is also important to note that all of the data that authors present are based on the KO/OE of the proteins both of which, as the authors also demonstrate, cause significant changes in the expression level for hundreds of genes, many of which are RBPs. The manuscript could therefore be strengthened by the further validation of direct RNA-protein interaction by LIS1. Related comments and suggestions are detailed below.

Major point

1. Page 5, line 107

Further explanation on the relationship between LIS1 and the metabolites such as fatty acids would be helpful and graphical and statistical analysis of the differentially expressed metabolites could be provided to enhance the understanding of the data.

In addition to Supplementary Table 3, we have added Supplementary Figure 2.

Supplementary Figure 2: LIS1 dosage affects the cellular metabolome. A) Metabolomic analysis of mESCs with varying LIS1 levels. F/- vs. WT: Metabolites of *Lis1* flox/- (F/-) compared to that extracted from wildtype (WT). F/- vs. OE: F/- metabolites plotted against LIS1-GFP overexpressing line (OE). OE vs. WT: Metabolites extracted from LIS1-GFP overexpressing line compared against WT metabolome. B) Comparative fatty acid metabolic profile of the same lines. The p-values are color-coded.

2. Page 7, line 143 "LIS1 interactome" and Fig. 2a-c.

Based on the result and the figure, it is not completely clear to me how the authors determined LIS1 interactome, which they claim to be 726 proteins. Could the result be presented in similar manner to that of the AGO 1-4KO or Dox.AGO2: IP-LIS1 mass spectrometry analysis result, Figure 1. f-g?

We have added the LIS1 interacting proteins for each comparison (Supplementary Table 5A-C) and included additional interacting proteins from BioGRID in Supplementary Table 5C. The data presented in the AGO 1-4KO or Dox.AGO2: IP-LIS1 mass spectrometry analysis result contains

most of the proteins detected using IP from the different cell lines. We compiled a list of the LIS1 interactome resulting in 1274 proteins.

Modified text:

The LIS1 interactome

The known LIS1 interactome is composed of 148 proteins, many of which are involved in microtubule-based processes (BioGRID⁸ and STRING⁹ databases), and very few are nuclear proteins. Novel LIS1-interacting proteins were identified by immunoprecipitation of LIS1 from nuclear or cytoplasmic fractions of mESCs expressing graded levels of LIS1 followed by mass-spectrometry (LC-MS/MS) (Fig. 2A, Supplementary Fig. 4A-D). A total of 726 unique proteins were identified, many of which were not known to complex with LIS1 (Supplementary Table 5A, Supplementary Fig. 4E-F). More than half of these proteins (385) were designated in the RBPbase (<https://rbpbase.shiny.embl.de/>) as "RNA binding-GO-Mm". Most of the LIS1 interactome was shared in the cells with different LIS1 expression levels (WT, F/-, and OE) (Supplementary Fig. 4E). The top consistent and genotype-specific GO terms related to mRNA splicing and cellular response to stress (Supplementary Fig. 4F). Many LIS1-interacting proteins belonged to the RISC complex and P-bodies, interacting with AGO2 (Fig. 2B). A second prominent group of proteins was composed of RNA-binding proteins functioning in other activities, including alternative splicing, mRNA stabilization, RNA metabolism, transcriptional and translational regulation (Fig. 2C). We found several members of the heterogeneous nuclear ribonucleoproteins (hnRNPs)¹⁰, small nuclear ribonucleoproteins (snRNPs)¹¹, DDX, and DHX gene families¹². A subset of these two groups of proteins is shown in the heatmaps, demonstrating that the protein-protein interactions are enriched in the nucleus and occur in cells with graded levels of LIS1. We composed a list of the LIS1 interactome by combining our data with BioGRID data resulting in 1274 unique proteins (Supplementary Table 5A-C).

More importantly, based on the manuscript including the methods section, it seems that IP experiment was performed without nuclease treatment. Especially since the LIS1 is now considered as a RBP, it would be interesting to perform the same experiment with and without nuclease treatment to distinguish/exclude the RNA dependent interaction from that of the protein-protein interactions, as it is commonly done for the RPB IP-MS studies (**Li et al. 2016**)

We observed a strong interaction with AGO2 in co-immunoprecipitation with and without benzonase nuclease treatment to remove all RNA/DNA in Figure 2 F.

The text was modified:

We confirmed the interaction between LIS1 and AGO2 by co-immunoprecipitation followed by western blot in the nuclear and cytoplasmic fractions (Fig. 2E). We treated protein lysates with benzonase nuclease and observed that LIS1 immunoprecipitated AGO2, suggesting that LIS1 and AGO2 interact regardless of the presence of RNA in the complex (Fig. 2 F).

3. Page 9, line 190 "previous high throughput studies identified LIS1 as an RBP"

LIS1 does not harbor any of the known classical RBDs but is one of the WD40 repeat proteins, which was implicated in RNA binding activity (Jin et al. 2016, Castello et al. 2012). It would be more convincing if the authors could further elaborate on the basis of RNA-protein interaction conducted by LIS1 including if the protein is identified in **more than one RNA interactome profiling studies, apart from the one the authors cited, conducted in the mouse and human cells or other model organisms (Hentze et al. 2018, Panhale et al. 2019, Na et al. 2021).**

We checked the datasets and elaborate below on the detection of LIS1 in MS.

LIS1 was detected as an RNA-binding protein in *Xenopus*¹³ only using the FAX-RIC method both in oocytes and embryos. Was not detected in^{14, 15, 16, 17, 18, 19, 20, 21}. Was detected as an RNA-binding protein in *Drosophila* using CAPRI (Crosslinked and Adjacent Peptides-based RNA-binding domain Identification)^{22, 23}. LIS1 was detected in mouse embryonic stem cells⁷ (previously cited) and in yeast²⁴.

The related text was modified:

An additional survey of the literature demonstrated that LIS1 was detected as an RBP in additional species²³ including the clawed frog¹³, fruit flies²², and yeast²⁴, highlighting this evolutionarily conserved property.

4. Page 9, line 203

Could the LIS1-RNA interaction have specificity to the GU-rich sequence? If so further experimental validation or data analysis to show that LIS1's sequence dependent RNA interactions, which is independent of the fact that LIS1 binds the donor splice site, would serve as a compelling evidence to indicate direct RNA binding activity of LIS1.

We have performed de novo motif analysis in addition to the known motif and found consistent specificity to GU-rich sequence, updated in Figure 3 C.

5. Figure 5h

It isn't clear to me why the authors chose not to perform similar comparison with the **wild type and/or LIS1 KO or OE cells**. The authors suggested that AGO 1-4 KO **"likely" alter the physical properties of the cells and does not provide the data supporting that assumption**. It is also puzzling that Dox.AGO2 & LIS1-OE resulted in **slightly lowered elastic modulus**, in other words no definitive conclusion can be drawn from this result, other than the fact that LIS1 OE can affect the stiffness of mESCs in one way or another, which is rather expected based on the known function of LIS1 protein.

We added Gene analytics enrichment to compare LIS1 OE And Dox AGO2 from total RNA seq data.

In addition, we see that differential expression of miR-302 and miR-221 family is involved in pluripotency, differentiation, and changes in hippo signaling, thus regulating mechanosensitivity in ESCs.

We have performed a western blot for YAP in AGO1-4 KO lines. We see that an increase in LIS1 GFP can independently modulate the levels of YAP in the absence of AGO2. On comparing the levels of YAP, we suggest that LIS1 and AGO2 may have diverging pleiotropic effects on mechanosensitive pathways.

Modified text:

A potential factor that may affect the mechanosensitivity of the ESCs is YAP. We found that LIS1 OE and/or AGO2 introduction significantly increased YAP expression (Fig. 6F,G).

Added to the discussion:

In addition, LIS1 OE in the absence of AGO1-4 resulted in increased YAP expression, which is involved in mechanotransduction pathways.

Minor point

1. Figure 1d. and other related figure.

Color code and the size of dots are hard to distinguish and the meaning of number need to be explained on the figure legend.

We corrected it.

2. Page 4, line 87, Page 5, line 97 “(Fig. 1B)” – Inconsistent letter case.

We corrected it.

References

Jin W, Wang Y, Liu CP, et al. Structural basis for snRNA recognition by the double-WD40 repeat domain of Gemin5. *Genes Dev.* 2016;30(21):2391-2403. doi:10.1101/gad.291377.116

Castello A, Fischer B, Eichelbaum K, et al. Insights into RNA biology from an atlas of mammalian mRNA-binding proteins. *Cell.* 2012;149(6):1393-1406. doi:10.1016/j.cell.2012.04.031

Yang Li, Mahlon Collins et al. Immunoprecipitation and mass spectrometry defines an extensive RBM45 protein–protein interaction network, *Brain Research*, Volume 1647, 2016, Pages 79-93, ISSN 0006-8993, <https://doi.org/10.1016/j.brainres.2016.02.047>.

Panhale, A., Richter, F.M., Ramírez, F. et al. CAPRI enables comparison of evolutionarily conserved RNA interacting regions. *Nat Commun* 10, 2682 (2019). <https://doi.org/10.1038/s41467-019-10585-3>

Na Y, Kim H, Choi Y, et al. FAX-RIC enables robust profiling of dynamic RNP complex formation

in multicellular organisms in vivo. *Nucleic Acids Res.* 2021;49(5):e28. doi:10.1093/nar/gkaa1194

Hentze MW, Castello A, Schwarzl T, Preiss T. A brave new world of RNA-binding proteins. *Nat Rev Mol Cell Biol.* 2018;19(5):327-341. doi:10.1038/nrm.2017.130

Reviewer #3 (Remarks to the Author):

In the manuscript titled "LIS1 RNA-binding orchestrates the mechanosensitive properties of embryonic stem cells in AGO2-dependent and independent ways", the authors used multiple high-throughput methods to characterize how different doses of LIS1 impact post-transcriptional regulation in embryonic stem cells (ESCs). The authors uncovered several interesting properties of (nuclear) LIS1 such as its interaction with AGO2 and many RNA-binding proteins (RBPs) and its possible regulatory role in RNA splicing and microRNA (miRNA) expression. With the potential caveat of LIS1 overexpression, it would have been an impactful work if the authors could have provided a clear example on how, at the physiological level, LIS1 participates post-transcriptional regulation in ESCs with mechanistic information. However, the manuscript is mostly descriptive and in some part lacks a coherent narrative regarding the biological functions of LIS1 in post-transcriptional regulation. Overall, the manuscript in its current form unfortunately is not a strong candidate to be considered for publication in *Nature Communications*.

Nevertheless, the current work does raise several interesting questions: What are the (strongest) effects of LIS1 on gene expression (including mRNAs and small RNAs)? How does it affect gene expression mechanistically (direct/indirect)? How does this previously unknown role in gene expression fit into the known phenotype from LIS1 mutations? Those are not easy questions but if the authors could provide insights into some of the aforementioned questions, possibly through additional biological investigations guided by the existing multi-modality data, it will surely elevate the impact from the current work.

#Main comments:

1) p.4~6, Paragraph: Dosage sensitive effects of LIS1 in pluripotent networks: What's the expression level of LIS1 OE in both mouse and human ESC? From the Methods section, CAG promoter, a strong promoter, is used to drive LIS1-GFP expression. I understand that the authors set out to investigate the LIS1 dosage effects and rescue the lethal effects from LIS1 deletion. However, LIS1-OE mESC are used to investigate effects on gene expression in many of the following sections, and supra-physiologic level of LIS-GFP expression can have effects on gene expression that are not present at the endogenous level. **The authors need to perform western blot to directly compare LIS1/LIS1-GFP expression and estimate the dosage of LIS1 across all the cell-lines used. This will provide an important reference to the discovered effects on gene expression.**

We have performed western blots and quantified the levels of LIS1 for all the lines. For LIS1-GFP OE (mouse and human) Figure 1 E and Supplementary Fig 1C (mouse) and Supplementary Fig. 3A, B (human).

2) p.4~6, Paragraph: Dosage sensitive effects of LIS1 in pluripotent networks: Is it possible to provide a comparison of the DE genes between mESC and hESC (with comparable dosage effects)?

Approximately 150 genes are common between mouse and human differentially expressed genes. We have now added an additional sheet to Supplementary Table 4.

Modified in the text:

A subset of the differentially expressed genes was common to mice and humans (Supplementary Table 4C).

3) p.9, Extended Data Fig. 3b/c:

-The comparison is to genome, what about unbound (but expressed) genes (i.e., non-target)? Is it still significant?

This relates to the eCLIP data. When compared to the genome, it is significant $p < 0.05$ (supplementary Fig. 5B). When compared to unbound genes, we do not see a significant difference.

However, in the case of miRs, we see a significant difference, as detailed in the text and Figure 4 E, statistics in Supplementary Table 7 C.

In the text:

We noted that the expression of miRs close to LIS1 seCLIP peaks is significantly higher than those located at a distance from the LIS1 seCLIP peaks (Fig. 4D,E). We then examined the expression of miRs in relation to LIS1 seCLIP peaks using a threshold of 2kb (Fig. 4E, statistics in Supplementary Table 7C). The expression of a miR located in the LIS1 seCLIP site (± 2 kb) was significantly higher than in all other categories. If the location of the miR was further away from the peak (> 2 kb), their level of expression was dependent on whether they were in non-coding or protein-coding introns, with those located in introns of non-coding RNAs having higher expression than those found in protein-coding introns or in comparison to all other miRs (statistics in Supplementary Table 7C).

-I know the information is provided in the supplemental table, but it will help to have the p-value on the plot as well.

We added the p-values to Supplementary Figures 5B,C.

We have added to Figure 4 E part of the comparisons. Details of the statistics of all possible comparisons are in Supplementary Table 7C.

4) p.9~10, Paragraph: LIS1 is an RNA-binding protein: Are the mESC RNA-seq data presented in this paragraph (OE, WT, F/-) new or are they part of the RNA-seq data described in Fig. 1c? Please clarify in the text. If they are not new, please use the same exact designation (including 4-OHT usage) from Fig. 1c.

They are new cell lines generated from blastocysts collected from the respective mice. We added to the text:

To better understand the impact of LIS1-RNA binding, we sequenced *Lis1* F/-, WT, and LIS1-OE mouse ESC lines generated from blastocysts.

5) p.9, "...We detected a negative correlation between the number of LIS1 seCLIP sites and splicing efficiency in wild-type RNAseq data". Were Lefty2 and Cdh1 splicing efficiency affected? If yes, please add the RNA-seq data to Fig. 3d/e to show the splicing phenotype. If no, would it be possible to provide other examples to show that LIS1 intronic binding affects splicing efficiency of the target genes?

A complete list of transcripts showing changes in splicing efficiency and bound to LIS1 are outlined in Supplementary Table 8 D.

Lefty2 and Cdh1 splicing do not seem to be affected but rather their expression.

Added to the text:

A complete list of transcripts showing significant changes in splicing efficiency and bound to LIS1 are outlined in Supplementary Table 8D.

6) p.10, "...Retained intron events were plotted, showing more events occurring in the F/-". Does this mean there are more intron retention events in F/- compared with OE?

It indicates that reduced levels of LIS1 contribute to higher intron retention events (Fig. 3G).

Added text:

These results indicate that when LIS1 levels are reduced, splicing efficiency decreases, leading to increased intron retention events.

If LIS1 binding is negatively correlated with splicing efficiency, isn't the expectation to be MORE intron retention in OE? Please clarify and explain.

The splicing efficiency and the number of LIS1 binding clusters are correlated with the levels of LIS1.

Added to the text:

These results indicate that when LIS1 levels are reduced, splicing efficiency decreases, leading to increased intron retention events. Conversely, in the case of LIS1 OE, it is possible to note fewer events of intron retention (Fig. 3G), suggesting that the splicing out of introns from the nascent transcripts is more efficient. As shown above, elevated LIS1 levels result in more efficient splicing than F/- within the group of genes with the same LIS1 seCLIP binding sites (Fig. 3F).

7) p.9~10, Paragraph: LIS1 is an RNA-binding protein:

-The authors need to fully integrate of LIS1 seCLIP data with RNA-seq data to identify how direct LIS1 binding affects splicing in candidate genes. Does direct LIS1 binding block or promote splicing? Just like other RBPs, LIS1 can do both in different genes but the authors **need to parse their data further to find out.**

We compared the list of genes that exhibit differential splicing efficiency between OE and F/- (Table 8D and the DE OE and F/-) with the alternatively spliced genes in the same comparison detected by MAJIQ and found 1251 genes (added new Table 8E). For several of the genes (highlighted in yellow) we placed MAJIQ outputs. Regarding the question of whether LIS1 binding blocks or promotes splicing, it really depends on the specific transcript.

Added to the text:

We further compared the list of genes that exhibited differential expression and differential splicing efficiency (OE and F/-) with the alternatively spliced genes detected by MAJIQ, resulting in 1251 genes (Table 8E).

-One caveat about existing data: it is a good choice to perform LIS1-seCLIP in wildtype mESC with physiological level LIS1 expression. However, overexpression may result in additional binding not seen at physiologic conditions. If LIS1 is expressed a lot higher under LIS1-OE condition, the observed splicing phenotype from LIS1-OE can be caused by binding not captured in the existing LIS1-seCLIP data.

We agree with this comment, and it may explain the above.

Modified in the text:

In addition, we wish to note that in the case of the LIS1 OE, additional LIS1 binding sites may not have been captured in our LIS1 seCLIP data set derived from WT cells.

8) p.9~10, Paragraph: LIS1 is an RNA-binding protein: Is there any splicing difference detected between WT and F/-? The information should also be provided even if there are very few

changes, as the disease phenotype, lissencephaly, is caused by LIS1 loss-of-function rather than overexpression.

We have performed differential splicing analysis using MAJIQ and RMATS to compare WT and F/-. The data relating to significant differences have been in Supplementary Table 6 D-F.

Added to the text:

We detected a negative correlation between the number of LIS1 seCLIP sites and splicing efficiency in wild-type RNAseq data (Fig. 3F, green, Bulk RNA-Seq splicing data using MAJIQ and RMATS in Supplementary Table 6, and the statistics in Supplementary Table 7B). The most marked changes were noted when comparing *LIS1 OE* to *Lis1 F/-* (Supplementary Table 6A-C), and fewer events were noted when comparing the WT to *Lis1 F/-* (Supplementary Table 6D-F).

9) p. 11~12, Paragraph: LIS1 affects the expression of miRs: Same as 8). What's the rationale focusing on the DE list from LIS1 overexpression? In Fig. 4a, it looks like there is a group of miRs express lower in F/- compared to WT and OE (middle). The authors need to comment on this group of miRs as well.

miR302a-3p, miR302b-3p, miR302c-3p, miR302d-3p, miR302a-5p, miR-142a-3p, miR-335-3p, miR-335-5p, miR-211-5p. These are the miRNAs common between OE and WT versus F/-. Previous studies have shown that miR302 cluster is important in mechanosensitivity, neural differentiation, and reprogramming^{25, 26, 27, 28, 29, 30}.

Added to the text:

Another smaller group of miRs exhibited lower expression in the F/- compared to WT and OE. These miRs included miR302a-3p, miR302b-3p, miR302c-3p, miR302d-3p, miR302a-5p, miR-142a-3p, miR-335-3p, miR-335-5p, and miR-211-5p. Previous studies have shown that miR302 cluster is important in mechanosensitivity, neural differentiation, and reprogramming^{25, 26, 27, 28, 29, 30}.

10) p.12~13, Paragraph: LIS1 affects the expression of miRs without the Argonaute complex: It is very intriguing that LIS1 can upregulate a subset of miRs in the absence of Ago2. miR expression is complex and involves multiple steps. However, it will elevate the impact of the manuscript if the authors can provide any example as to how LIS1 affects miR expression in the absence of Ago2 mechanistically.

We agree with the reviewer that this is a very intriguing possibility.

Added to the text:

To explore the possible mechanism of the increased expression of miRs in the absence of all Argonaute proteins, we treated the cells with an RNA polymerase II inhibitor (alpha-amanitin)

and tested the expression of *Rian* and miR-314 (Supplementary Fig. 9B). Whereas the steady-state levels of *Rian* decreased in both cell lines, as expected, miR-341 decreased in the AGO 1-4 KO cells, but was stable in the LIS1 OE line suggesting that in some cases LIS1 may stabilize selected miRs.

Additional section in the discussion:

Therefore, our data suggest that LIS1 is involved in the positive regulation of the expression of these miRs possibly by binding and stabilizing them as indicated by the inhibition of RNA polymerase II experiment (Supplementary Fig. 9B). LIS1 might be involved in the recruitment of additional proteins to the nascent RNA or affecting RNA splicing involved in the formation of those miRs³¹. However, other mechanisms can contribute to these DE miRs.

#Minor comments:

1) Fig. 3h legend: Please define upregulation / downregulation: which genotype relative to which genotype?

It is defined now.

2) Extended Data Fig. 3a: What antibody was used for this western blot? The same anti-LIS1 used for IP? If the labels are correct, the western blot didn't quite show a clean IP with LIS1 only.

What are the additional bands in the IP lanes? Since LIS1 does interact with many other RBPs, how do you make sure there is only LIS1 in your IP?

We have used mouse anti-LIS1 (338) monoclonal antibody. Probably this is just a technical artifact for this experiment. We see a very clear band in other experiments, including IP for AGO2, and it is also knockout verified, as shown in Figure 2 E and Figure 1 E, respectively.

3) Extended Data Fig. 4d/e: Please provide more explanation for the various panels (especially panels at the bottom row): It's not clear what are the difference in splicing between F/- and OE from the busy plots.

We have now added sashimi plots to Supplementary Figure 6 D for a simpler representation of the difference in splicing from the *Meg3* locus. Detailed transcript-specific splice events are in Supplementary Table 6.

Added to the text:

An example of differential splicing between the F/- and the OE is shown using Sashimi plots for the *Meg3* locus (Supplementary Fig. 6 D).

1. Guruharsha KG, *et al.* A protein complex network of *Drosophila melanogaster*. *Cell* **147**, 690-703 (2011).
2. Sarshad AA, *et al.* Argonaute-miRNA Complexes Silence Target mRNAs in the Nucleus of Mammalian Stem Cells. *Mol Cell* **71**, 1040-1050 e1048 (2018).
3. Lessel D, *et al.* Germline AGO2 mutations impair RNA interference and human neurological development. *Nat Commun* **11**, 5797 (2020).
4. Bays JL, Campbell HK, Heidema C, Sebbagh M, DeMali KA. Linking E-cadherin mechanotransduction to cell metabolism through force-mediated activation of AMPK. *Nat Cell Biol* **19**, 724-731 (2017).
5. Aban CE, *et al.* Downregulation of E-cadherin in pluripotent stem cells triggers partial EMT. *Sci Rep* **11**, 2048 (2021).
6. Redmer T, Diecke S, Grigoryan T, Quiroga-Negreira A, Birchmeier W, Besser D. E-cadherin is crucial for embryonic stem cell pluripotency and can replace OCT4 during somatic cell reprogramming. *EMBO Rep* **12**, 720-726 (2011).
7. He C, *et al.* High-Resolution Mapping of RNA-Binding Regions in the Nuclear Proteome of Embryonic Stem Cells. *Mol Cell* **64**, 416-430 (2016).
8. Stark C, Breitkreutz BJ, Reguly T, Boucher L, Breitkreutz A, Tyers M. BioGRID: a general repository for interaction datasets. *Nucleic Acids Res* **34**, D535-539 (2006).
9. Szklarczyk D, *et al.* The STRING database in 2021: customizable protein-protein networks, and functional characterization of user-uploaded gene/measurement sets. *Nucleic Acids Res* **49**, D605-D612 (2021).
10. Geuens T, Bouhy D, Timmerman V. The hnRNP family: insights into their role in health and disease. *Hum Genet* **135**, 851-867 (2016).
11. Stone LB, Riley KJ. Small Nuclear Ribonucleoproteins (snRNPs). In: *eLS* (2014).
12. Abdelhaleem M, Maltais L, Wain H. The human DDX and DHX gene families of putative RNA helicases. *Genomics* **81**, 618-622 (2003).
13. Na Y, *et al.* FAX-RIC enables robust profiling of dynamic RNP complex formation in multicellular organisms in vivo. *Nucleic Acids Res* **49**, e28 (2021).
14. Baltz AG, *et al.* The mRNA-bound proteome and its global occupancy profile on protein-coding transcripts. *Mol Cell* **46**, 674-690 (2012).

15. Castello A, *et al.* Insights into RNA biology from an atlas of mammalian mRNA-binding proteins. *Cell* **149**, 1393-1406 (2012).
16. Castello A, *et al.* Comprehensive Identification of RNA-Binding Proteins by RNA Interactome Capture. *Methods Mol Biol* **1358**, 131-139 (2016).
17. Beckmann BM, *et al.* The RNA-binding proteomes from yeast to man harbour conserved enigmRBPs. *Nat Commun* **6**, 10127 (2015).
18. Kramer K, *et al.* Photo-cross-linking and high-resolution mass spectrometry for assignment of RNA-binding sites in RNA-binding proteins. *Nat Methods* **11**, 1064-1070 (2014).
19. Conrad T, Albrecht AS, de Melo Costa VR, Sauer S, Meierhofer D, Orom UA. Serial interactome capture of the human cell nucleus. *Nat Commun* **7**, 11212 (2016).
20. Queiroz RML, *et al.* Comprehensive identification of RNA-protein interactions in any organism using orthogonal organic phase separation (OOPS). *Nat Biotechnol* **37**, 169-178 (2019).
21. Trendel J, *et al.* The Human RNA-Binding Proteome and Its Dynamics during Translational Arrest. *Cell* **176**, 391-403 e319 (2019).
22. Panhale A, *et al.* CAPRI enables comparison of evolutionarily conserved RNA interacting regions. *Nat Commun* **10**, 2682 (2019).
23. Hentze MW, Castello A, Schwarzl T, Preiss T. A brave new world of RNA-binding proteins. *Nat Rev Mol Cell Biol* **19**, 327-341 (2018).
24. Scherrer T, Mittal N, Janga SC, Gerber AP. A screen for RNA-binding proteins in yeast indicates dual functions for many enzymes. *PLoS One* **5**, e15499 (2010).
25. Subramanyam D, *et al.* Multiple targets of miR-302 and miR-372 promote reprogramming of human fibroblasts to induced pluripotent stem cells. *Nat Biotechnol* **29**, 443-448 (2011).
26. Lin SL, Chang DC, Lin CH, Ying SY, Leu D, Wu DT. Regulation of somatic cell reprogramming through inducible mir-302 expression. *Nucleic Acids Res* **39**, 1054-1065 (2011).
27. Parchem RJ, *et al.* miR-302 Is Required for Timing of Neural Differentiation, Neural Tube Closure, and Embryonic Viability. *Cell Rep* **12**, 760-773 (2015).

28. Yang SL, Yang M, Herrlinger S, Liang C, Lai F, Chen JF. MiR-302/367 regulate neural progenitor proliferation, differentiation timing, and survival in neurulation. *Dev Biol* **408**, 140-150 (2015).
29. Xu F, *et al.* MicroRNA-302d promotes the proliferation of human pluripotent stem cell-derived cardiomyocytes by inhibiting LATS2 in the Hippo pathway. *Clin Sci (Lond)* **133**, 1387-1399 (2019).
30. Tian Y, *et al.* A microRNA-Hippo pathway that promotes cardiomyocyte proliferation and cardiac regeneration in mice. *Science translational medicine* **7**, 279ra238 (2015).
31. Flynt AS, Greimann JC, Chung WJ, Lima CD, Lai EC. MicroRNA biogenesis via splicing and exosome-mediated trimming in *Drosophila*. *Mol Cell* **38**, 900-907 (2010).

REVIEWERS' COMMENTS

Reviewer #1 (Remarks to the Author):

In this revision Kshrisagar et al. have added new gene expression analyses and other data in response to the comments. While the authors have addressed some of the previous comments, the functional role of the novel Lis1 RNA-binding property in regulating ES cell biology and/or cell differentiation and survival is still somewhat poorly defined.

Major Comments

1. To confirm that the Lis1 mediated regulation of splicing (and RNA-binding) is of functional significance, the authors should determine if over-expressing any of the identified downstream genes can functionally rescue the impact of Lis1 loss on cell death and/or ECM related gene expression. The data showing that downstream gene expression patterns with Lis1 loss correlating with DNA damage, apoptosis etc. (Supp Table 8C) is not sufficient to conclusively establish a functional role for the novel Lis1 RNA-binding property.
2. Given that the gene expression analysis is driving several conclusions of functional impact of Lis1 loss in ES cells, it would be helpful to represent the several GSEA analysis in main figures as dot plots that highlight the top significantly enriched pathways with gene scores in Lis1 WT and KO/OE cells (and not just as multiple spreadsheets Table 4b, 13B, 8C).

Reviewer #2 (Remarks to the Author):

I appreciate the additional experiments and the efforts taken by the authors to address the comments. Additional figure and analysis improved the manuscript significantly. Demonstration of sequence motif specific and functionally relevant interaction of LIS1 and RNA transcripts seem to strongly indicate that there is novel post-transcriptional regulation related function of LIS1 which is the main hypothesis and the strength of this study.

The manuscript is also in good shape except for the few typos that I noted below.

Page 16 line 354 "revealed the that" typo

Page 20 line 438 "the passenger stands of the" typo

Reviewer #3 (Remarks to the Author):

All my comments/concerns are satisfactorily addressed in the revised manuscript and I would like to recommend publication.

Sunday, April 30, 2023

Final revisions for Nature Communications manuscript NCOMMS-22-12510A

A point-by-point response to the reviewers' comments

We want to thank the anonymous reviewers for taking the time and effort to review our manuscript and to improve it.

Please find below our response.

Reviewer #1 (Remarks to the Author):

In this revision Kshrisagar et al. have added new gene expression analyses and other data in response to the comments. While the authors have addressed some of the previous comments, the functional role of the novel Lis1 RNA-binding property in regulating ES cell biology and/or cell differentiation and survival is still somewhat poorly defined.

Major Comments

1. To confirm that the Lis1 mediated regulation of splicing (and RNA-binding) is of functional significance, the authors should determine if over-expressing any of the identified downstream genes can functionally rescue the impact of Lis1 loss on cell death and/or ECM related gene expression. The data showing that downstream gene expression patterns with Lis1 loss correlating with DNA damage, apoptosis etc. (Supp Table 8C) is not sufficient to conclusively establish a functional role for the novel Lis1 RNA-binding property.

We thank the reviewer for pointing this out. We have tried to overexpress AGO2 to test if it can functionally rescue the effects of LIS1 loss on RNA-splicing. Our results indicate that the overexpression of AGO2 had no effect.

We believe that additional rescue experiments are beyond the scope of this manuscript.

2. Given that the gene expression analysis is driving several conclusions of functional impact of Lis1 loss in ES cells, it would be helpful to represent the several GSEA analysis in main figures as dot plots that highlight the top significantly enriched pathways with gene scores in Lis1 WT and KO/OE cells (and not just as multiple spreadsheets Table 4b, 13B, 8C).

We added related dot plots, due to space limitations we added two to supplementary figures and one to a main figure.

The data that was presented in Table 4b (currently Supplementary Data 3b) is shown in Supplementary Fig. 3d.

The data that was presented in Table 8C (currently Supplementary Data 7c) is shown in Supplementary Fig. 7b.

The data that was presented in Table 13B (currently Supplementary Data 12b) is shown in Fig. 6c.

Reviewer #2 (Remarks to the Author):

I appreciate the additional experiments and the efforts taken by the authors to address the comments. Additional figure and analysis improved the manuscript significantly. Demonstration of sequence motif specific and functionally relevant interaction of LIS1 and RNA transcripts seem to strongly indicate that there is novel post-transcriptional regulation related function of LIS1 which is the main hypothesis and the strength of this study.

The manuscript is also in good shape except for the few typos that I noted below.

Page 16 line 354 “revealed the that” typo

Page 20 line 438 “the passenger stands of the” typo

We thank the reviewer for noticing these typos and we corrected them in the current version.

Reviewer #3 (Remarks to the Author):

All my comments/concerns are satisfactorily addressed in the revised manuscript and I would like to recommend publication.

Thank you!